# On the Optimal Construction of Unbiased Gradient Estimators for Zeroth-Order Optimization

**Shaocong Ma**
Department of Computer Science
University of Maryland
College Park, MD 20742, USA
scma0908@umd.edu

**Heng Huang**\*
Department of Computer Science
University of Maryland
College Park, MD 20742, USA
heng@umd.edu

## Abstract

Zeroth-order optimization (ZOO) is an important framework for stochastic optimization when gradients are unavailable or expensive to compute. A potential limitation of existing ZOO methods is the bias inherent in most gradient estimators unless the perturbation stepsize vanishes. In this paper, we overcome this biasedness issue by proposing a novel family of *unbiased* gradient estimators based solely on function evaluations. By reformulating directional derivatives as a telescoping series and sampling from carefully designed distributions, we construct estimators that eliminate bias while maintaining favorable variance. We analyze their theoretical properties, derive optimal scaling distributions and perturbation stepsizes of four specific constructions, and prove that SGD using the proposed estimators achieves optimal complexity for smooth non-convex objectives. Experiments on synthetic tasks and language model fine-tuning confirm the superior accuracy and convergence of our approach compared to standard methods.

## 1 Introduction

In this paper, we consider the problem of *zeroth-order optimization (ZOO)*, where our goal is to solve the following stochastic optimization problem:

$$\min_{x \in \mathbb{R}^d} f(x) := \mathbb{E}_{\xi \sim \Xi} f(x; \xi), \tag{1}$$

where $f(x; \xi)$ is a smooth loss function evaluated on data $\xi$ drawn from a distribution $\Xi$. In many practical scenarios, gradient information is either unavailable or prohibitively expensive to compute. Due to its versatility, ZOO has been widely adopted across various domains, including black-box adversarial attacks on machine learning models [Chen et al., 2017, Kurakin et al., 2016, Papernot et al., 2017, Cai et al., 2021, Zhao et al., 2020], physics-informed neural networks interfacing with external PDE solvers [Shen et al., 2024, Ma et al., 2025], and reinforcement learning [Choromanski et al., 2018, Lei et al., 2022, Suh et al., 2022]. Recent research on ZOO also focuses on enhancing memory efficiency [Cai et al., 2022a,b, Li et al., 2024, Sugiura and Matsutani, 2025], motivated in large part by fine-tuning large language models [Malladi et al., 2023, Zhang et al., 2024, Gautam et al., 2024, Tang et al., 2024, Wang et al., 2024, 2025].

Unlike first-order methods that rely on stochastic gradients $\nabla f(x; \xi)$, ZOO uses only function evaluations, without access to gradient information. To approximate gradients, several estimators have been proposed, including the one-point estimate $\hat{\nabla} f(x; \xi) = \frac{f(x + \mu v; \xi)}{\mu} v$ [Flaxman et al., 2005, Shamir, 2013, Bach and Perchet, 2016, Nesterov and Spokoiny, 2017, Berahas et al., 2022] and

---

\*This work was partially supported by NSF IIS 2347592, 2348169, DBI 2405416, CCF 2348306, CNS 2347617, RISE 2536663.

two-point estimator $\hat{\nabla} f(x; \xi) = \frac{f(x+\mu v; \xi) - f(x; \xi)}{\mu} v$ [Ghadimi and Lan, 2013, Duchi et al., 2015, Nesterov and Spokoiny, 2017] (see Appendix A.1 for further discussions). The random direction $v$ is typically drawn from a Gaussian or uniform spherical distribution, while alternative choices have also gained increasing attention in recent years [Ghadimi and Lan, 2013, Duchi et al., 2015, Ji et al., 2019, Sahu et al., 2019, Coope and Tappenden, 2020, Kozak et al., 2023, Rando et al., 2024a,b, Ma and Huang, 2025, Mi et al., 2025].

However, despite these advancements, a critical limitation arises in zeroth-order gradient estimation; that is, all widely used gradient estimators exhibit inherent bias. Specifically, unless the perturbation step size $\mu$ asymptotically tends to zero, these estimators yield persistently biased approximations of the true gradient. This inherent bias motivates a central question explored in this paper:

> ***Q1**: Is it possible to design an unbiased zeroth-order gradient estimator using only function evaluations?*

**Contribution 1**: In this paper, we answer **Q1** affirmatively. Contrary to the belief that zeroth-order gradient estimators must inherently be biased due to finite-step perturbations, we demonstrate that it is indeed possible to construct *unbiased* gradient estimators using only function evaluations. Our key idea is to express $\nabla_v f(x)$ (in the deterministic setting), the directional derivative along the direction $v$, as a telescoping series:

$$
\begin{aligned}
\nabla_v f(x) &:= \lim_{\mu_n \to 0} \frac{f(x + \mu_n v) - f(x)}{\mu_n} \\
&= \sum_{n=1}^{\infty} p_n \left[ \frac{f(x + \mu_1 v) - f(x)}{\mu_1} + \frac{1}{p_n} \left( \frac{f(x + \mu_{n+1} v) - f(x)}{\mu_{n+1}} - \frac{f(x + \mu_n v) - f(x)}{\mu_n} \right) \right], \quad (2) \\
&\stackrel{(i)}{=} \mathbb{E}_{n \sim \{p_n\}_{n=1}^{\infty}} \left[ \frac{f(x + \mu_1 v) - f(x)}{\mu_1} + \frac{1}{p_n} \left( \frac{f(x + \mu_{n+1} v) - f(x)}{\mu_{n+1}} - \frac{f(x + \mu_n v) - f(x)}{\mu_n} \right) \right]
\end{aligned}
$$

where the perturbation stepsize $\mu_n \to 0$ as $n \to \infty$, the sampling distribution $\{p_n\}_{n=1}^{\infty}$ form a probability distribution (that is, $0 < p_i < 1$ for all $i \in \mathbb{N}$ and $\sum_{i=1}^{\infty} p_i = 1$), and the expectation representation (i) holds under mild regularity conditions (Proposition 2.1). This formulation allows us to reinterpret the directional derivative as an expectation over $n \sim \{p_n\}_{n=1}^{\infty}$, enabling the construction of a unbiased gradient estimator family $\mathscr{P}$ (Definition 2.2):

$$
\hat{\nabla}_v f(x) := \mathbb{E}_{n \sim p} \mathsf{P}(n, v),
$$

where $\mathsf{P}(n, v)$ is an unbiased estimator of

$$
\frac{f(x + \mu_1 v) - f(x)}{\mu_1} + \frac{1}{p_n} \left( \frac{f(x + \mu_{n+1} v) - f(x)}{\mu_{n+1}} - \frac{f(x + \mu_n v) - f(x)}{\mu_n} \right).
$$

Within this framework, we propose four specific estimators, denoted as $\mathsf{P}_k$-*estimator* for $k = 1, 2, 3, 4$, corresponding to the number of function evaluations required in each estimation. To the best of our knowledge, unbiased zeroth-order gradient estimators have received little attention in prior literature. The only existing work we are aware of is the four-point estimator proposed by Chen [2020], which shares the same telescoping structure and can be viewed as a special case of our $\mathsf{P}_4$-estimator.

**Contribution 2:** Building on our unbiased estimator construction, we conduct a rigorous variance analysis on our proposed $\mathsf{P}_k$-estimators. We first present a negative result for the $\mathsf{P}_1$-estimator; although it requires fewer function evaluations, it may exhibit infinite variance under certain conditions (Theorem 3.1 (a)), which aligns with the one-point estimator in the randomized smoothing [Flaxman et al., 2005]. Next, we characterize the relation among the variance of the $\mathsf{P}_k$-estimator ($k = 2, 3, 4$), the perturbation stepsize sequence $\{\mu_n\}_{n=1}^{\infty}$, and the sampling distribution $\{p_n\}_{n=1}^{\infty}$ (Theorem 3.1 (b)). Identifying the optimal choice of $\{\mu_n\}_{n=1}^{\infty}$ and $\{p_n\}_{n=1}^{\infty}$ leads us to the following *non-convex functional optimization* problem:

$$
\min_{\{\mu_n\}_{n=1}^{\infty}, \{p_n\}_{n=1}^{\infty}} \quad \mathbb{E}_{n \sim \{p_n\}_{n=1}^{\infty}} \left( \frac{\mu_n - \mu_{n+1}}{p_n} \right)^2 \tag{3}
$$

$$
\text{subject to} \quad 0 < p_n < 1; \ \sum_{n=1}^{\infty} p_n = 1; \ \sum_{n=1}^{\infty} \mu_n < \infty.
$$

We present an explicit analytical solution to this optimization problem (Theorem 3.2), which reveals two key insights: (1) our constructed unbiased gradient estimators can achieve the same variance as the classical two-point estimator without introducing additional bias, leading to the best-possible complexity for SGD algorithm (Corollary 3.5); (2) a broad class of sampling distributions can achieve the minimum variance, extending beyond the specific choices considered in prior work [Chen, 2020]. While our theoretical results establish strong guarantees, an important practical question remains:

> **Q2:** Given the optimal choice of $\{\mu_n\}_{n=1}^{\infty}$ and $\{p_n\}_{n=1}^{\infty}$, do the proposed unbiased estimators empirically outperform existing zeroth-order methods?

**Contribution 3:** To address **Q2**, we empirically validate our proposed approach across both synthetic and practical tasks. On estimating the gradient of mean-square and logistic losses, our method achieves significantly lower gradient estimation error compared to standard zeroth-order methods (Section 4.1). Furthermore, when applied to fine-tuning large language models, the proposed estimators demonstrate faster convergence and higher final accuracy under the same number of function evaluations (Section 4.2). These results confirm the practical advantages of our unbiased construction and underscore its effectiveness in modern zeroth-order optimization tasks.

## 2 The Derivation of Unbiased Zeroth-Order Estimators

We will start from the deterministic case then turn to the stochastic case in Section 3.3. In this section, we formally derive a class of unbiased estimators for approximating the gradient $\nabla f(x)$ using only function evaluations. We also provide a sufficient condition under which the telescoping series in Eq. (4) admits the expectation representation. All proofs are provided in the appendix.

### 2.1 Telescoping Series and Expectation Representation

For a fixed direction $v \in \mathbb{R}^d$, the directional derivative of a differentiable function $f : \mathbb{R}^d \to \mathbb{R}$ at $x$ along the direction $v$ is defined as

$$\nabla_v f(x) = \lim_{\mu \to 0} \frac{f(x + \mu v) - f(x)}{\mu}.$$

Then for any decreasing sequence $\{\mu_n\}_{n=1}^{\infty}$ with $\lim_{n \to \infty} \mu_n = 0$, one can express this directional derivative as the limit of a convergent sequence $\left\{ \frac{f(x+\mu_n v) - f(x)}{\mu_n} \right\}$:

$$\nabla_v f(x) = \lim_{n \to \infty} \frac{f(x + \mu_n v) - f(x)}{\mu_n}.$$

This convergent sequence canonically induces a telescoping series with the same limit:

$$\nabla_v f(x) = \frac{f(x + \mu_1 v) - f(x)}{\mu_1} + \sum_{n=1}^{\infty} \left[ \frac{f(x + \mu_{n+1} v) - f(x)}{\mu_{n+1}} - \frac{f(x + \mu_n v) - f(x)}{\mu_n} \right]. \quad (4)$$

Next, consider a probability mass function (PMF) $\{p_n\}_{n=1}^{\infty}$ with $p_n > 0$ for all $n$ and $\sum_{n=1}^{\infty} p_n = 1$. When the series in Eq. (4) is ***absolutely convergent***[2], we can interpret it as an expectation over a discrete random variable $n$. That is,

$$
\begin{aligned}
\nabla_v f(x) &= \sum_{n=1}^{\infty} p_n \left[ \frac{f(x + \mu_1 v) - f(x)}{\mu_1} + \frac{1}{p_n} \left( \frac{f(x + \mu_{n+1} v) - f(x)}{\mu_{n+1}} - \frac{f(x + \mu_n v) - f(x)}{\mu_n} \right) \right] \\
&= \mathbb{E} \left[ \frac{f(x + \mu_1 v) - f(x)}{\mu_1} + \frac{1}{p_n} \left( \frac{f(x + \mu_{n+1} v) - f(x)}{\mu_{n+1}} - \frac{f(x + \mu_n v) - f(x)}{\mu_n} \right) \right].
\end{aligned}
$$
(5)

---

[2]We follow the standard definition from Spivak [2008]: A series $\sum_{n=1}^{\infty} a_n$ is called *convergent*, if the limit of its finite sum $\lim_{N \to \infty} \sum_{n=1}^{N} a_n$ exists. A series $\sum_{n=1}^{\infty} a_n$ is called *absolutely convergent*, if the series $\sum_{n=1}^{\infty} |a_n|$ is convergent. See the formal definition in Appendix B.1.

**On the Role of Absolute Convergence.** The absolute convergence of the series in Eq. (4) plays a critical role in interpreting the telescoping series as an expectation. This is due to the difference between the series convergence and the existence of expectation:

- *The series convergence*: Consider the convergent series $\sum_{i=1}^{\infty} p_i x_i$. To evaluate its value, we can calculate the finite-sum $S_n := \sum_{i=1}^{n} p_i x_i$; then we have $\sum_{i=1}^{\infty} p_i x_i = \lim_{n \to \infty} S_n$.

- *The existence of expectation*: Consider the random variable $X$ with $\mathbb{P}(X = x_i) = p_i$ for $i \in \mathbb{N}$. Its expectation $\mathbb{E}[X]$ is also written as $\sum_{i=1}^{\infty} p_i x_i$. However, the notion of expectation must be well-defined independently of any ordering of outcomes. That is, for an arbitrary permutation $\sigma : \mathbb{N} \to \mathbb{N}$, all series $\sum_{i=1}^{\infty} p_{\sigma(i)} x_{\sigma(i)}$ should represent the same value $\mathbb{E}[X]$.

As a result, a convergent series can yield different values depending on the order of summation (this result is called the Riemann series theorem [Riemann, 1868, Spivak, 2008]); however, the outcomes of a random variable requires a random variable's expectation to be well-defined regardless of any such ordering. While the expectation representation has been discussed in prior work (e.g., [Chen, 2020]), the lack of attention to absolute convergence has left the conditions ensuring unbiasedness underexplored.

Due to this reason, we provide the following (mild) sufficient condition for ensuring the absolute convergence with adding a slightly stronger requirement on the objective function $f : \mathbb{R}^d \to \mathbb{R}$ and the sequence $\{\mu_n\}_{n=1}^{\infty}$:

**Proposition 2.1.** *If the second-order continuously differentiable function $f : \mathbb{R}^d \to \mathbb{R}$ has L-Lipschitz continuous gradient and $\sum_{n=1}^{\infty} \mu_n < \infty$, then the series*

$$\sum_{n=1}^{\infty} p_n \left[ \frac{f(x + \mu_1 v) - f(x)}{\mu_1} + \frac{1}{p_n} \left( \frac{f(x + \mu_{n+1} v) - f(x)}{\mu_{n+1}} - \frac{f(x + \mu_n v) - f(x)}{\mu_n} \right) \right]$$

*is absolutely convergent and its limit is $\nabla_v f(x)$.*

## 2.2 The Construction of Unbiased Estimators

With the expectation representation in place, we are ready to define the class of unbiased estimators explicitly.

**Definition 2.2.** Suppose that the function $f : \mathbb{R}^d \to \mathbb{R}$ is continuously differentiable and $\{\mu_n\}_{n \geq 1}$ is a positive sequence with $\lim_{n \to \infty} \mu_n = 0$ such that the telescoping series

$$\frac{f(x + \mu_1 v) - f(x)}{\mu_1} + \sum_{n=1}^{\infty} \left[ \frac{f(x + \mu_{n+1} v) - f(x)}{\mu_{n+1}} - \frac{f(x + \mu_n v) - f(x)}{\mu_n} \right]$$

is absolutely convergent, the sequence $\{p_n\}_{n=1}^{\infty}$ forms a PMF, and $V$ is the distribution over $\mathbb{R}^d$. Then the family of estimators $\mathscr{P} := \mathscr{P}(f, \{\mu_n\}_{n=1}^{\infty}, \{p_n\}_{n=1}^{\infty}, V)$ denote the class of random variables such that for every $\mathsf{P}(n, v) \in \mathscr{P}$, it satisfies

$$\mathbb{E}[\mathsf{P}(n, v) \mid n, v] = \frac{f(x + \mu_1 v) - f(x)}{\mu_1} + \frac{1}{p_n} \left( \frac{f(x + \mu_{n+1} v) - f(x)}{\mu_{n+1}} - \frac{f(x + \mu_n v) - f(x)}{\mu_n} \right),$$

where $v$ is sampled from $V$, independent with $n \sim \{p_n\}_{n=1}^{\infty}$.

In the following theorem, we formally prove that our proposed class $\mathscr{P}$ is exactly the unbiased estimator of the gradient $\nabla f(x)$.

**Theorem 2.3** (Unbiasedness). *Let $\mathscr{P} := \mathscr{P}(f, \{\mu_n\}_{n=1}^{\infty}, \{p_n\}_{n=1}^{\infty}, V)$ is defined as Definition 2.2. Then, for any estimator $\mathsf{P}(n, v) \in \mathscr{P}$, the following hold:*

*(a) $\mathbb{E}[\mathsf{P}(n, v) \mid v] = \nabla_v f(x)$; that is, $\mathsf{P}(n, v)$ is an unbiased estimator of the directional derivative $\nabla_v f(x)$.*

*(b) If the random direction $v$ is chosen independently of the sampling $n \sim \{p_n\}_{n=1}^{\infty}$ and satisfies $\mathbb{E}[v v^\top] = I$, then*

$$\mathbb{E}_{n \sim \{p_n\}_{n=1}^{\infty}, v \sim V} \left[ \mathsf{P}(n, v) v \right] = \nabla f(x),$$

*so that $\mathsf{P}(n, v) v$ is an unbiased estimator of the gradient $\nabla f(x)$.*

## 2.3 Specific Constructions

In this subsection, we propose four concrete constructions from the estimator family (Definition 2.2)

$$\mathscr{P} := \mathscr{P}(f, \{\mu_n\}_{n=1}^{\infty}, \{p_n\}_{n=1}^{\infty}, V)$$

based on the number of function evaluations used in estimating the gradient. These constructions are designed to explore two main aspects: (1) the trade-off between the estimator variance and the number of function evaluations, allowing flexibility depending on the computational budget; and (2) a fundamental question purely driven by the theoretical interest: *What is the minimum number of function evaluations required to construct an unbiased gradient estimator?*

$\mathsf{P}_4$-**Estimator.** This estimator corresponds to the four-point estimator originally proposed by Chen [2020] with slightly generalizing the choice of the perturbation stepsize sequence $\{\mu_n\}_{n=1}^{\infty}$ and the sampling distribution $\{p_n\}_{n=1}^{\infty}$. For a given direction $v \sim V$ and $n \sim \{p_n\}_{n=1}^{\infty}$, the $\mathsf{P}_4$-estimator is defined as

$$\mathsf{P}_4(n, v) = \frac{f(x + \mu_1 v) - f(x)}{\mu_1} + \frac{1}{p_n}\left[\frac{f(x + \mu_{n+1} v) - f(x)}{\mu_{n+1}} - \frac{f(x + \mu_n v) - f(x)}{\mu_n}\right]. \quad (6)$$

This construction requires four function evaluations at: $x$, $x + \mu_1 v$, $x + \mu_n v$, and $x + \mu_{n+1} v$, exhibiting the lowest variance and the most function evaluation counts among all members of $\mathscr{P}$.

$\mathsf{P}_3$-**Estimator.** We can reduce one function evaluation by introducing a selection random variable $\mathsf{U}_2 \sim \mathrm{Uniform}(\{0, 1\})$[3]. The estimator is then defined as

$$\mathsf{P}_3(n, v) = \frac{f(x + \mu_1 v) - f(x)}{\mu_1}\mathsf{U}_2 + \frac{1}{p_n}\left[\frac{f(x + \mu_{n+1} v) - f(x)}{\mu_{n+1}} - \frac{f(x + \mu_n v) - f(x)}{\mu_n}\right](1 - \mathsf{U}_2). \quad (7)$$

This construction randomly selects one of two pathways: With probability $1/2$, it uses the first term only and requires two function evaluations at $x + \mu_1 v$ and $x$; otherwise, it uses the second term and requires three function evaluations at $x + \mu_n v$, $x + \mu_{n+1} v$, and $x$. This estimator maintains unbiasedness as $\mathsf{P}_4$, with slightly higher variance.

$\mathsf{P}_1$- & $\mathsf{P}_2$-**Estimator.** The selection random variable can be naturally extended to construct $\mathsf{P}_1$- and $\mathsf{P}_2$-estimators as follows:

$$\mathsf{P}_2(n, v) = \frac{f(x + \mu_1 v) - f(x)}{\mu_1}\mathbb{I}_{\{\mathsf{U}_3=0\}} \quad (8)$$
$$+ \frac{1}{p_n}\left[\frac{f(x + \mu_{n+1} v) - f(x)}{\mu_{n+1}}\mathbb{I}_{\{\mathsf{U}_3=1\}} - \frac{f(x + \mu_n v) - f(x)}{\mu_n}\mathbb{I}_{\{\mathsf{U}_3=2\}}\right],$$

$$\mathsf{P}_1(n, v) = \frac{f(x + \mu_1 v)\mathbb{I}_{\{\mathsf{U}_4=1\}} - f(x)\mathbb{I}_{\{\mathsf{U}_4=0\}}}{\mu_1} \quad (9)$$
$$+ \frac{1}{p_n}\left[\frac{f(x + \mu_{n+1} v)\mathbb{I}_{\{\mathsf{U}_4=2\}} - f(x)\mathbb{I}_{\{\mathsf{U}_4=0\}}}{\mu_{n+1}} - \frac{f(x + \mu_n v)\mathbb{I}_{\{\mathsf{U}_4=3\}} - f(x)\mathbb{I}_{\{\mathsf{U}_4=0\}}}{\mu_n}\right],$$

where $\mathsf{U}_3 \sim \mathrm{Uniform}(\{0, 1, 2\})$, $\mathsf{U}_4 \sim \mathrm{Uniform}(\{0, 1, 2, 3\})$, and $\mathbb{I}_A$ is the indicator function, which equals 1 if the event $A$ occurs, and 0 otherwise. of the event $A$. Remarkably, the construction of $\mathsf{P}_1$-estimator achieves unbiasedness using only a single function evaluation. However, we will show that in the next section, $\mathsf{P}_1$-estimator will have infinite variance under certain condition.

## 3 Variance Analysis of Unbiased Zeroth-Order Estimators

In this section, we provide a theoretical analysis of the variance behavior for the unbiased estimator family $\mathscr{P} = \mathscr{P}(f, \{\mu_n\}_{n=1}^{\infty}, \{p_n\}_{n=1}^{\infty}, V)$ (Definition 2.2). While the unbiasedness has been shown

---

[3]Here we use $\mathrm{Uniform}(A)$ to represent the uniform distribution over the finite or compact set $A$.

in Theorem 2.3, their variances can differ dramatically depending on the estimator construction. In particular, we prove that the variance becomes unbounded (i.e., infinite) for certain constructions such as $P_1$-estimator. We also provide finite-variance bounds for $P_k$-estimators (for $k = 2, 3, 4$ with matching the optimal variance under specific choices of $\{p_n\}$ and $\{\mu_n\}$.

## 3.1 Theoretical Analysis

In the following result, we adopt the same condition as Proposition 2.1 to ensure the expectation representation.

**Theorem 3.1.** *Let $\mathscr{P} := \mathscr{P}(f, \{\mu_n\}_{n=1}^{\infty}, \{p_n\}_{n=1}^{\infty}, V)$ is defined as Definition 2.2. Suppose that $f : \mathbb{R}^d \to \mathbb{R}$ is second-order continuously differentiable and has L-Lipschitz continuous gradient, $\sum_{n=1}^{\infty} \mu_n < \infty$, and $V$ is the uniform distribution over the sphere with the radius $\sqrt{d}$. Define*

$$\mu := \mu_1, \qquad \varrho := \sum_{n=1}^{\infty} \frac{(\mu_{n+1} - \mu_n)^2}{p_n}, \qquad \text{and} \qquad \varphi := \sum_{n=1}^{\infty} \frac{\mu_n^2}{p_n}.$$

*Then the following statements hold:*

(a) *If there exists a point $x \in \mathbb{R}^d$ such that the Hessian $\nabla^2 f(x)$ is positive definite and $f(x) \neq 0$, then the variances of the $P_1$ for estimating $\nabla f(x)$ is infinite.*

(b) *The variance of $P_k$-estimator $P_k(n, v)v$ ($k = 2, 3, 4$) for estimating $\nabla f(x)$ is given by*

$$\text{Var}[P_2(n, v)\, v] \leqslant \text{Var}[P_4(n, v)\, v] + \frac{L^2}{3} d^3 \mu^2 + \frac{L^2}{12} d^3 \varrho + \frac{L^2}{3} d^3 \varphi.$$

$$\text{Var}[P_3(n, v)v] \leqslant \text{Var}[P_4(n, v)\, v] + \frac{L^2}{8} d^3 \mu^2 + \frac{L^2}{8} d^3 \varrho.$$

$$\text{Var}[P_4(n, v)v] \leqslant (d - 1)\|\nabla f(x)\|^2 + \frac{3L^2}{4} d^3 \mu^2 + \frac{L^2 d^3}{2} \varrho.$$

*Proof.* Part (a) directly follows by analyzing the tail of $\frac{1}{p_n} \frac{f(x + \mu_n v)}{\mu_n}$ and leveraging the curvature from a positive definite Hessian. For the part (b), we simply decompose the variance of $P_2(n, v)v$ and $P_3(n, v)v$ into the variance of estimating $P_4(n, v)v$ using

$$\text{Var}[P\, v] = d\mathbb{E}[(P - P_4(n, v))^2] + \text{Var}[P_4(n, v)\, v]$$

for arbitrary $P := P(n, v) \in \mathscr{P}$. Then we apply the second-order Taylor expansions with the mean value theorem to control the finite-difference noise. Full details and auxiliary lemmas are provided in Appendix C. □

**Comparison with Existing Literature.** Theorem 3.1 reveals that while $P_1$ is unbiased, its variance can be infinite under certain conditions, making them unsuitable for SGD. In contrast, $P_k$-estimator ($k = 2, 3, 4$) offer the finite variance when $\{\mu_n\}_{n=1}^{\infty}$ and $\{p_n\}_{n=1}^{\infty}$ are appropriately selected. We will show it later that under the optimal setting, their variances match the optimal order of classical two-point estimators [Nesterov and Spokoiny, 2017] but with zero bias:

$$\text{Var}[P_k\, v] = \mathcal{O}(d\|\nabla f(x)\|^2 + d^3 \mu^2).$$

This variance will lead to the optimal function query complexity $\mathcal{O}(\frac{d}{\epsilon^4})$ for achieving $\epsilon$-accuracy in the gradient norm $\|\nabla f(x)\|$ [Duchi et al., 2015].

**Comparison with the Noisy Oracle Setup** In our work, we consider the exact function evaluation setting with noiseless values. In this case, our variance scales as $d^3 \mu^2$, which is worse than the $d^2 \mu^2$ of some specific biased estimators, which is mitigated by choosing a small enough $\mu$; the overall sample complexity remains optimal. However, in the noisy function evaluation setting, where each function evaluation may return a noisy value, a smaller $\mu$ amplifies the noise, leading to degraded performance. Several recent works have provided more refined analysis under noisy setups with improved variance behavior. Notably, Akhavan et al. [2024] demonstrated that for highly smooth functions, the $\ell_1$-randomization can reduce the variance scaling to $d^2 \mu^2$ with achieving the improved

performance for highly smooth objective functions, which extends the existing $\ell_1$-randomization proposed by Akhavan et al. [2022]. Earlier work by Gasnikov et al. [2017] analyzed the variance behavior in single-point and multi-point bandit feedback settings, and more recent developments further explore the impact of first-order smoothness in noisy black-box optimization [Gasnikov et al., 2022]. Notably, all of these results achieve the optimal complexity derived by Duchi et al. [2015].

## 3.2 On the Optimal Choices of $\{\mu_n\}_{n=1}^{\infty}$ and $\{p_n\}_{n=1}^{\infty}$

In previous section, Theorem 3.1 connects the perturbation stepsize sequence $\{\mu_n\}$, the sampling distribution $\{p_n\}$, and the variance upper bounds of our constructed unbiased estimators, which has received limited discussion in the existing literature. To control the variance term, one must ensure that $\varrho := \sum_{n=1}^{\infty} \frac{(\mu_{n+1} - \mu_n)^2}{p_n}$ (and $\varphi := \sum_{n=1}^{\infty} \frac{\mu_n^2}{p_n}$ for $\mathsf{P}_2$-estimator) is sufficiently small. This observation naturally raises the question: What are the optimal sequences $\{\mu_n\}_{n=1}^{\infty}$ and $\{p_n\}_{n=1}^{\infty}$ that minimize this sum? The following theorem addresses this question:

**Theorem 3.2.** *Let $\{\mu_n\}_{n=1}^{\infty}$ be a positive, decreasing sequence with $\sum_{n=1}^{\infty} \mu_n < \infty$, and let $\{p_n\}_{n=1}^{\infty}$ be a PMF. Denote $\mu := \mu_1$. Then the following statements hold:*

(a) *The lower bound of $\varrho$ is given by $\varrho \geqslant \mu^2$. Moreover, the equality holds if and only if $p_n = \frac{\mu_n - \mu_{n+1}}{\mu}$.*

(b) *The lower bound of $\varphi$ is given by $\varphi \geqslant \left( \sum_{n=1}^{\infty} \mu_n \right)^2 > \mu^2$. Moreover, the equality holds if and only if $p_n = \frac{\mu_n}{\sum_{n=1}^{\infty} \mu_n}$.*

This result characterizes the choices of $\{\mu_n\}_{n=1}^{\infty}$ and $\{p_n\}_{n=1}^{\infty}$ that minimizes $\varrho$ (and $\varphi$ for the $\mathsf{P}_2$-estimator), leading to the variance upper bound of the form:

$$\max\{\mathrm{Var}[\mathsf{P}_2(n,v)v], \mathrm{Var}[\mathsf{P}_3(n,v)v], \mathrm{Var}[\mathsf{P}_4(n,v)v]\} \leqslant \mathcal{O}(d\|\nabla f(x)\|^2 + d^3\mu^2).$$

Here, we can always choose $\{\mu_n\}_{n=1}^{\infty}$ for the $\mathsf{P}_2$-estimator such that $\mu_1 \approx \sum_{n=2}^{\infty} \mu_n$ to nearly match the lower bound ($\approx \mu_1^2$).

**Sampling from the Optimal Sampling Distribution $\{p_n\}_{n=1}^{\infty}$.** When the perturbation stepsize sequence $\{\mu_n\}_{n=1}^{\infty}$ is given, sampling the corresponding optimal distribution $p_n = \frac{\mu_n - \mu_{n+1}}{\mu_1}$ could be difficult; in most of cases, $\{p_n\}_{n=1}^{\infty}$ cannot be a ready-to-use distribution naively supported by existing software. Fortunately, we can do it conversely: given an arbitrary PMF $\{p_n\}_{n=1}^{\infty}$, the perturbation stepsize takes the form

$$\mu_n = \mu_1 \mathbb{P}(N \geqslant n), \quad \text{where } N \sim \{p_n\}_{n=1}^{\infty},$$

providing a practical way to implement the unbiased zeroth-order gradient estimator. To illustrate this point, we provide two concrete examples.

**Example 3.3** (Geometric $\mathsf{P}_k$-Estimators)**.** We consider the geometric distribution $n \sim \mathrm{Geom}(c)$ ($c \in (0,1)$). Then $p_n = (1-c)c^{n-1}$ for all $n \in \mathbb{N}$. We define $\mu_n$ by the recursion $\mu_n - \mu_{n+1} = \mu_1 p_n = \mu_1(1-c)c^{n-1}$. Summing this relation leads to the closed-form solution

$$\mu_n = \mu_1 c^{n-1}$$

and the optimal value $\varrho = \mu_1^2$. This construction recovers the geometric sampling scheme used by Chen [2020]. We call the $\mathsf{P}_k$-estimator constructed on the geometric distribution as the *geometric $\mathsf{P}_k$-estimator*. It is easy to verify that the corresponding $\varphi$ is given as $\varphi = \sum_{n=1}^{\infty} \frac{\mu_n^2}{p_n} = \frac{\mu_1^2}{(1-c)^2}$.

**Example 3.4** (Zipf's $\mathsf{P}_k$-Estimators)**.** We consider the Zipf's distribution $n \sim \mathrm{Zipf}(s)$ ($s > 1$). Then $p_n = \frac{1}{\zeta(s)} \frac{1}{n^s}$ for all $n \in \mathbb{N}$, where $\zeta$ is the Riemannian zeta function defined as $\zeta(s) = \sum_{n=1}^{\infty} \frac{1}{n^s}$. We define $\mu_n$ by the recursion $\mu_n - \mu_{n+1} = \mu_1 p_n = \mu_1 \frac{1}{\zeta(s)} \frac{1}{n^s}$. Summing this relation leads to the closed-form solution

$$\mu_n = \mu_1 \left[ 1 - \frac{\sum_{j=1}^{n-1} \frac{1}{j^s}}{\zeta(s)} \right].$$

This construction also leads to the optimal value $\varrho = \mu_1^2$. When estimating the upper bound of $\varphi$, we additionally assume $s > 3$. In this case, we have $\varphi = \sum_{n=1}^{\infty} \frac{\mu_n^2}{p_n} \leqslant \frac{\zeta(s-2)}{(s-1)^2 \zeta(s)} \mu_1^2$. The detailed calculation is put in Example C.7.

In both examples, we start with a well-known easy-to-sample distribution $\{p_n\}_{n=1}^{\infty}$, and calculate the associated perturbation stepsize sequence $\{\mu_n\}_{n=1}^{\infty}$ either analytically (Geometric $\mathsf{P}_k$-estimators) or iteratively (Zipf's $\mathsf{P}_k$-estimators). While all estimators (i.e. $\mathsf{P}_k$-estimator with $k = 2, 3, 4$) achieve the optimal variance in the order with $d$ and $\mu$, these examples indicate a key difference between the $\mathsf{P}_2$-estimator and the $\mathsf{P}_k$-estimator (for $k = 3, 4$): the variance bound of $\mathsf{P}_3$- and $\mathsf{P}_4$-estimator is ***parameter-agnostic***; that is, once $\{p_n\}$ is specified, no additional tuning of distribution parameters is required to attain the optimal bound $\mu^2$. This distinction highlight the practical advantages of $\mathsf{P}_3$- and $\mathsf{P}_4$-estimators.

### 3.3 Convergence of SGD with Unbiased Gradient Estimators

In this subsection, we consider the stochastic optimization setting described in Eq. (1), where the goal is to estimate the stochastic gradient $\nabla f(x; \xi)$ rather than the full gradient. Under the optimal sampling distribution $\{p_n\}_{n=1}^{\infty}$ and the corresponding perturbation stepsize sequence $\{\mu_n\}_{n=1}^{\infty}$, the convergence upper bound of SGD follows directly from standard results for general unbiased stochastic gradient methods.

**Corollary 3.5** (Khaled and Richtárik [2022]). *Consider the stochastic optimization problem in Eq. (1), and suppose that the individual loss $f(x; \xi)$ is second-order differentiable with $L$-Lipschitz continuous gradient in $x$, uniformly over $\xi \sim \Xi$. Assume the stochastic gradient is approximated using the $\mathsf{P}_k$-estimator $\mathsf{P}_k(n, v) v$ for $k = 2, 3, 4$. Let the SGD iteration be defined as $x_{t+1} = x_t - \eta \mathsf{P}_k(n_t, v_t) v_t$ where $\eta \in (0, \frac{1}{L^2 d}]$ is the stepsize. Then the iterates satisfy the following convergence guarantee:*

$$\min_{0 \leqslant t \leqslant T-1} \mathbb{E}\|\nabla f(x_t)\|^2 \leqslant \mathcal{O}(d^3 \mu^2 \eta + d\eta + \frac{2}{\eta T}).$$

*Consequently, choosing $\eta = \Theta(1/\sqrt{dT})$ and $\mu = \mathcal{O}(\frac{1}{d})$ yields the optimal complexity $T = \Theta(\frac{d}{\epsilon^4})$ of having $\min_{0 \leqslant t \leqslant T-1} \mathbb{E}\|\nabla f(x_t)\| \leqslant \epsilon$.*

This complexity has matched the lower bound of solving a smooth non-convex optimization problem using zeroth-order gradient-based method [Duchi et al., 2015] and cannot be further improved without adding additional assumptions. Though we directly apply the result from Khaled and Richtárik [2022] (which is applicable for all unbiased estimators), the zeroth-order estimation can result in an additional dependence on the dimension $d$; this dependence has been reflected in our upper bound.

## 4 Experiments

To validate our theoretical results and demonstrate the effectiveness of the proposed unbiased zeroth-order gradient estimators, we conduct experiments on two settings: synthetic objectives and language model optimization. Details and hyperparameter configurations are provided in Appendix E.

### 4.1 Synthetic Examples

We first evaluate our estimators on two classic loss functions [James et al., 2013]: the quadratic loss $f_{\text{reg}} : \mathbb{R}^d \to \mathbb{R}$ for linear regression and the logistic loss $f_{\text{cls}} : \mathbb{R}^d \to \mathbb{R}$ for binary classification.

$$f_{\text{reg}}(x) = x^\top A^\top A x, \quad f_{\text{cls}}(x) = \frac{1}{n} \sum_{i=1}^{n} \log(1 + \exp(-b_i \cdot (a_i^\top \cdot x))),$$

where each entry of $A \in \mathbb{R}^{d \times d}$ is independently sampled from the uniform distribution $U[-1, 1]$, each feature vector $a_i \in \mathbb{R}^d$ is sampled from the standard normal distribution $\text{Normal}(0, I_d)$, and $b_i \in \{-1, 1\}$ are binary labels generated based on a Bernoulli distribution with the fixed sample size $n$. The gradient of each objective function can be explicitly evaluated; we compare the performance of different zeroth-order gradient estimator using the Mean-Square-Error (MSE), which is defined as

$$\text{MSE}(\hat{\nabla} f(x)) := [\hat{\nabla} f(x) - \nabla f(x)]^\top [\hat{\nabla} f(x) - \nabla f(x)]. \tag{10}$$

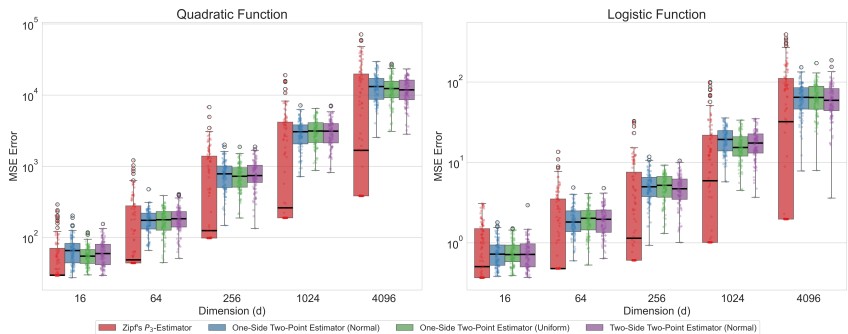

Figure 1: This figure presents the MSE error of four different estimators across various dimensions $d$ ranging from $16$ to $4096$. The left panel corresponds to the quadratic loss $f_{\text{reg}}$, while the right panel illustrates results for the logistic loss $f_{\text{cls}}$. Each box plot describes the distribution of the MSE error across $100$ random trials.

We compare the accuracy of estimating the gradient of two loss functions among four different gradient estimators including Zipf's $P_3$-estimator (Example 3.4), two-point estimator with Gaussian or uniform random perturbations, and centralized two-point estimator with uniform perturbation (the batch size of two-point estimators is adjusted to exactly $3$ function evaluations). For detailed hyper-parameter setting, we put in Appendix E. Several observations can be made from the results shown in Figure 1. First, comparing the same estimator across different dimensions, the MSE error for both objective functions generally increases with the dimension $d$, which is expected as higher-dimensional settings pose greater estimation challenges. Second, comparing different estimators, the Zipf's $P_3$-estimator consistently achieves lower MSE compared to others. These results collectively demonstrate the effectiveness of our proposed estimator when estimating the gradient, especially in high-dimensional settings, which will be further validate in the next experiment.

## 4.2 Language Model Optimization

In this section, we demonstrate the practical applicability of the unbiased gradient estimators in optimizing the deep neural network. Particularly, we apply it to the task of fine-tuning a pre-trained language model. Using zeroth-order optimization to fine-tune the LLMs has been an active research field in recent years due to its effectiveness in saving memory [Malladi et al., 2023, Zhang et al., 2024, Gautam et al., 2024, Guo et al., 2024]; it allows for fine-tuning model parameters without requiring access to the full computational graph, which can be prohibitively large for modern language models.

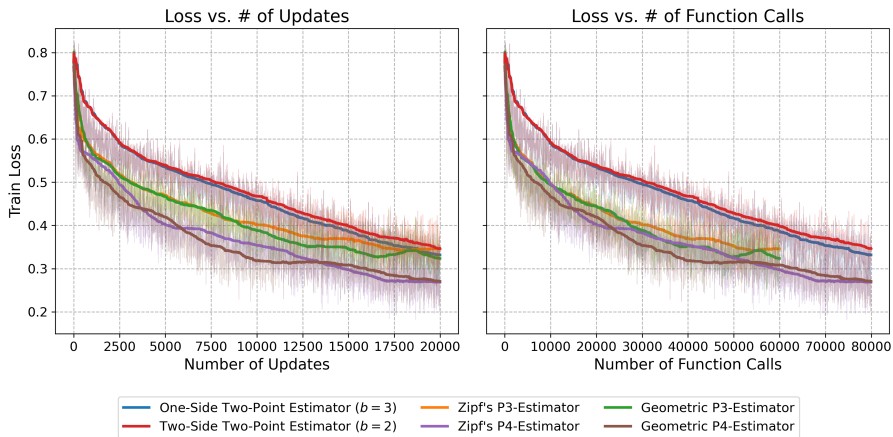

Figure 2: Comparison of training loss during fine-tuning of OPT-1.3B on SST-2 using different zeroth-order gradient estimators. The right panel rescales iterations by the number of function evaluations. The unbiased Zipf's $P_3$-, Zipf's $P_4$-, Geometric $P_3$-, and Geometric $P_4$-estimators achieve faster convergence under the same number of function evaluations.

We conducted experiments using the OPT-1.3b model [Zhang et al., 2022] for sentiment classification on the Stanford Sentiment Treebank (SST-2) dataset [Socher et al., 2013]. To ensure fair comparison, we maintained consistent parameters across experiments: the learning rate $\eta = 10^{-4}$ and the perturbation stepsize $\mu = 10^{-3}$ (corresponding to $\mu_1$ in the proposed unbiased estimators), which is taken from Malladi et al. [2023]'s Table 7 without additional tuning. For two-point estimators, we have adjusted the batch size to align 4 function evaluations. Detailed experimental settings are provided in Appendix E. As shown in Figure 2, zeroth-order optimization using the proposed unbiased zeroth-order estimators achieved superior performance compared to other baseline methods.

**Direct Comparison to the Two-Point Estimator ($b = 1$)** Previously, we compare our proposed methods against two-point estimators under the constraint of four function evaluations. It is also interesting to consider a direct comparison with classical two-point estimators using a batch size of $b = 1$, which corresponding to two function evaluations. Figure 3 presents this comparison.

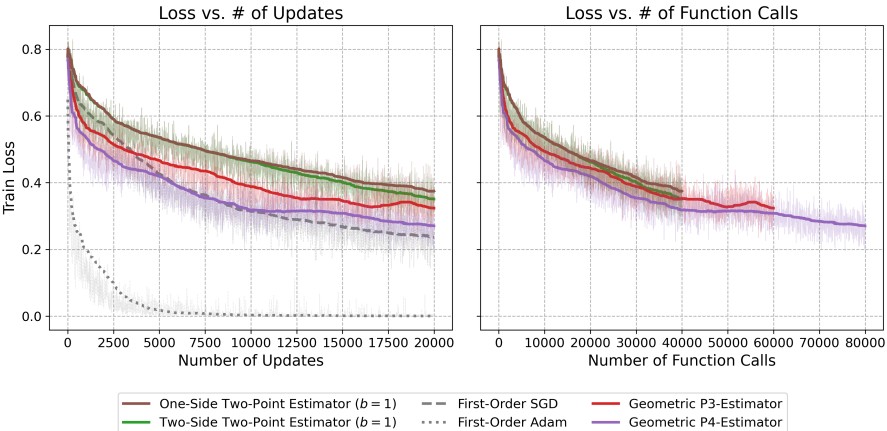

Figure 3: Comparison to the two-point estimator with $b = 1$ under the same setting as Figure 2. We also include the performance of the first-order Adam and SGD in the left panel.

Choosing larger batch sizes gives more accurate gradient estimates, leading to lower training loss when measured by the number of updates. However, we also observe that selecting the batch size as $b = 1$ may also present its own advantage. Therefore, choosing the batch size can be non-trivial and it requires to balance the variance of gradient estimation against the per-step cost.

## 5 Conclusion

In this work, we proposed a novel class of unbiased zeroth-order gradient estimators based on a telescoping series expansion of directional derivatives. We established new theoretical results, including a sufficient condition for the expectation representation (Proposition 2.1), the unbiasedness of the proposed estimators (Theorem 2.3), a variance analysis for four specific constructions (Theorem 3.1), and the characterization of the optimal sampling distribution and perturbation stepsize sequence (Theorem 3.2). We further demonstrated that SGD equipped with our estimators achieves optimal sample complexity and empirically outperforms existing mini-batch two-point estimators. These results provide a principled foundation for a new class of estimators in zeroth-order optimization, offering both theoretical insights and practical improvements.

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

# Appendix

## Table of Contents

## A  Additional Backgrounds

### A.1  Gradient Estimators in Zeroth-Order Optimization

**One-Point Zeroth-Order Estimator**    One-point estimators represent the simplest class, needing only a single function query per estimate. This construction makes them suitable when queries are costly or limited, like in online settings [Flaxman et al., 2005]. A common form, motivated by Gaussian smoothing [Nesterov and Spokoiny, 2017], is

$$\text{(Single-Point)} \qquad \hat{\nabla}_{\text{sgl}} f(x) = \frac{1}{\mu} f(x + \mu v) v,$$

where $v$ is often drawn from $\text{Normal}(0, I_d)$. While the expectation $\mathbb{E}[\frac{1}{\mu} f(x + \mu v) v]$ approximates the gradient of the smoothed function $\nabla_x \left[ \mathbb{E}_{v \sim \text{Normal}(0, I_d)} f(x + \mu v) \right]$, the estimator is biased regarding the true gradient $\nabla f(x)$. This bias diminishes as the smoothing parameter $\mu \to 0$ [Berahas et al., 2022]. However, these estimators suffer from high variance, potentially scaling with $d^2$ and exploding as $\mu \to 0$ [Flaxman et al., 2005].

**Two-Point Zeroth-Order Estimator**    Two-point estimators improve on one-point methods by using two function evaluations, often via a finite difference along a random direction [Shamir, 2017]. The standard difference form is

$$(\text{Two-Side}) \qquad \hat{\nabla}_{\text{2-side}}f(x) = \frac{f(x+\mu v) - f(x-\mu v)}{2\mu}v,$$

$$(\text{One-Side}) \qquad \hat{\nabla}_{\text{1-side}}f(x) = \frac{f(x+\mu v) - f(x)}{\mu}v,$$

requiring two queries [Shamir, 2017, Nesterov and Spokoiny, 2017]. This construction approximates the directional derivative [Chen, 2020]. Their expectation exactly matches the gradient of a smoothed function $\nabla_x f_\mu(x)$ [Nesterov and Spokoiny, 2017] and maintains a $\mathcal{O}(\mu)$-level error [Ma and Huang, 2025]. Variance is significantly lower than one-point methods, often scaling linearly with dimension $d$ [Duchi et al., 2015, Berahas et al., 2022].

**Multiple-Point Zeroth-Order Estimator**    Multiple-point estimators use more than two function evaluations to further enhance gradient estimate quality. Common strategies include Finite-Difference method [Dai, 2015] or mini-batch averaging [Duchi et al., 2015]. The finite-difference method approximates the gradient using finite differences along each standard basis direction:

$$(\text{Finite-Difference}) \qquad \hat{\nabla}_{\text{fin-diff}}f(x) = \sum_{i=1}^{d} \frac{f(x+\mu e_i) - f(x-\mu e_i)}{2\mu}e_i,$$

requiring $2d$ queries, where $e_i$ is the $i$-th coordinate vector. Mini-batch averaging reduces variance by averaging $b$ independent two-point estimates:

$$(\text{Mini-Batch}) \qquad \hat{\nabla}_{\text{batch}}f(x) = \sum_{i=1}^{b} \frac{f(x+\mu v_i) - f(x-\mu v_i)}{2\mu}v_i,$$

The finite-difference method offers low intrinsic variance but high query cost. Mini-batching reduces base estimator variance by $1/b$ at a cost of $b$ or $2b$ queries. These multi-point approaches can be combined arbitrary directional derivative estimators.

## A.2    Zeroth-Order SGD and Its Variants

**Vanilla Zeroth-Order SGD**    The convergence of SGD has been extensively studied under various settings. Ghadimi and Lan [2013] established complexity results for computing approximate solutions using first-order and zeroth-order (gradient-free) information with Gaussian smoothing. For smooth convex objective functions, Duchi et al. [2015] obtained the optimal convergence upper bound for SGD under the zeroth-order optimization (ZOO) setting. Nesterov and Spokoiny [2017] provided the optimal convergence upper bound for Gaussian smoothing. In the realm of nonconvex optimization, Ji et al. [2019] proposed two new zeroth-order variance-reduced optimization algorithms, ZO-SVRG-Coord-Rand and ZO-SPIDER-Coord, and provided improved analysis for the existing ZO-SVRG-Coord algorithm. These methods achieved better convergence rates and function query complexities than previous approaches. Berahas et al. [2022] derived convergence analyses for finite differences, linear interpolation, Gaussian smoothing, and uniform sphere smoothing methods. Recent studies have focused on non-smooth settings. Davis et al. [2022] and Zhang et al. [2020] established the sample complexity for Lipschitz functions without assuming smoothness. Lin et al. [2022] derived the complexity upper bound of SGD while noting a $\sqrt{d}$ scale compared to the smooth setting. Notably, Rando et al. [2024a] and Kornowski and Shamir [2024] revealed that by applying certain techniques, the non-smooth case is not inherently more challenging than the smooth case. A potential direction for extending this line of research is to explore the intersection between zeroth-order SGD and random reshuffling [Ma and Zhou, 2020, Mishchenko et al., 2020], minimax optimization [Chen et al., 2022], or dependent data [Ma et al., 2022].

**Variance-Reduced Zeroth-Order SGD**    A key bottleneck in vanilla zeroth-order SGD is the high variance of gradient estimators, which arises from both stochastic data sampling and the inherent randomness in the gradient estimation process. This high variance necessitates small stepsizes, leading to slow convergence [Liu et al., 2020]. To address the variance from stochastic data sampling,

variance reduction techniques—originally developed for first-order methods [Fang et al., 2018, Defazio et al., 2014, Johnson and Zhang, 2013, Nguyen et al., 2017], have been adapted to the zeroth-order setting. Algorithms such as ZO-SVRG [Liu et al., 2018, Huang et al., 2019, Gu et al., 2021], ZO-SVRG/SPIDER-Coord [Ji et al., 2019], and ZO-SPIDER/SARAH [Fang et al., 2018, Ji et al., 2019, Chen et al., 2023] leverage epoch-based updates with variance-reducing correction terms or recursive estimator refinements. These methods significantly improve convergence by reducing the iteration complexity.

**Memory-Efficient Zeroth-Order SGD**  Standard SGD typically requires storing all intermediate gradient across layers to enable chain-rule-based backpropagation, which incurs substantial memory overhead, especially when training large models. To alleviate this, MeZO [Malladi et al., 2023] introduces a memory-efficient approach wherein it suffices to store the random seed used to generate the perturbation vector for each layer, dramatically reducing memory consumption. This principle motivates algorithms such as Addax [Li et al., 2024], ElasticZO [Sugiura and Matsutani, 2025], and ZO2 [Wang et al., 2025], along with related benchmarking efforts [Zhang et al., 2024, Gautam et al., 2024, Wang et al., 2024, Guo et al., 2024]. Additional strategies exploit sparsity to further reduce memory usage; notable examples include ZORO [Cai et al., 2022b], the Extreme Sparsity framework [Guo et al., 2024], and the One-Bit method [Cai et al., 2022a].

## A.3   Discussions on the Forward Auto-Differentiation (AD) Approach

The forward gradient [Griewank and Walther, 2008] provides the exact directional derivative (with exactly zero bias), while the zeroth-order approach offers only an approximation of the derivative gradient. As a result, the zeroth-order approximation inherently introduces additional variance (even if it can be unbiased). As pointed out by Zhang et al. [2024], this makes the Forward AD method theoretically better in terms of estimator quality. However, there still multiple scenarios where the zeroth-order method is preferable.

- *Implementation Difficulty*: The practical implementation of Forward AD heavily relies on the availability of JVP (a.k.a. the Jacobian-Vector Product). A naive implementation will not reduce the memory usage and potentially increase the computation cost.
- *Memory usage*: Forward AD can be memory-efficient when implemented properly. However, it still presents a higher memory usage than the zeroth-order optimization. Therefore, for the edge device or other extreme cases where the memory cost is sensitive, we may still prefer the zeroth-order approach.

We also note that zeroth-order optimization is clearly advantageous in black-box settings where the forward gradient is not available. Therefore, the forward auto-differentiation and zeroth-order approaches are not mutually exclusive, but complementary, depending on the feasibility and the device memory.

## A.4   Discussions on the Difference Between Our Results and Chen [2020]

Although the telescoping structure is the same as the one used in Chen [2020] as we have commented in the introduction section, we have developed more results to the unbiased zeroth-order gradient estimator beyond this telescoping structure:

1. *Identify when we can have an unbiased gradient estimator*: We identify the condition under which the telescoping structure admits a valid expectation representation (Proposition 2.1). This condition is critical for constructing unbiased estimators, but has not been established in Chen [2020].

2. *More general unbiased gradient estimator & Reduce the number of function evaluations from 4 to 1*: The estimator in Chen [2020] is a special case of our $\mathsf{P}_4$-estimator. Our framework extends beyond this, answering a fundamental theoretical question: *What is the minimal number of function evaluations needed to construct an unbiased gradient estimator?* We improve the known answer from 4 give by Chen [2020] to 1.

3. *Identify the necessary and sufficient condition of achieving the optimal variance:* More-over, one of our key focuses is identifying optimal parameter sequences $\{\mu_n\}$ and $\{p_n\}$ (Theorem 3.1), rather than proposing a specific estimator.

# B Bias Analysis

## B.1 Absolute Convergence

In this subsection, we derive a sufficient condition to guarantee the expectation representation of the telescoping series introduced in Eq. (5). This requires ensuring the series is absolutely convergent, a property essential for interpreting it as the expectation of a well-defined random variable. The following definitions are directly taken from Folland [2002]:

**Definition B.1** (Convergent series). A series $\sum_{n=1}^{\infty} a_n$ is said to be *convergent* if the sequence of partial sums $S_N = \sum_{n=1}^{N} a_n$ is a convergent sequence; that is, $\lim_{N \to \infty} S_N$ exists.

**Definition B.2** (Absolutely convergent series). A series $\sum_{n=1}^{\infty} a_n$ is said to be *absolutely convergent* if the series $\sum_{n=1}^{\infty} |a_n|$ is convergent.

The following classical result, known as the Riemann series theorem [Riemann, 1868, Spivak, 2008], highlights the necessity of absolute convergence when interpreting an infinite sum as a well-defined value.

**Theorem B.3** (Riemann Series Theorem). *Let $\sum_{n=1}^{\infty} a_n$ be a conditionally convergent series of real numbers (i.e., convergent but not absolutely convergent). Then, for any real number $r \in \mathbb{R}$, there exists a rearrangement $\sigma : \mathbb{N} \to \mathbb{N}$ such that $\sum_{n=1}^{\infty} a_{\sigma(n)} = r$. Moreover, there exist rearrangements such that the sum diverges to $+\infty$, $-\infty$, or fails to converge at all.*

This theorem underscores the critical distinction between conditional and absolute convergence: a convergent series may yield different values under different summation orders. However, the definition of expectation for a random variable does not permit such ambiguity, since the outcomes have no inherent order. To guarantee a well-defined expectation, absolute convergence is required: that is, for a random variable $X$ with outcomes $\{x_n\}$ and probabilities $\{p_n\}$, it must hold that $\mathbb{E}|X| = \sum_{n=1}^{\infty} |x_n| p_n < \infty$.

We now recap Proposition 2.1 describing the condition where the telescoping series is absolutely convergent, enabling a valid expectation representation.

**Proposition B.4.** *If the second-order continuously differentiable function $f : \mathbb{R}^d \to \mathbb{R}$ has $L$-Lipschitz continuous gradient and $\sum_{n=1}^{\infty} \mu_n < \infty$, then the series*

$$\sum_{n=1}^{\infty} p_n \left[ \frac{f(x + \mu_1 v) - f(x)}{\mu_1} + \frac{1}{p_n} \left( \frac{f(x + \mu_{n+1} v) - f(x)}{\mu_{n+1}} - \frac{f(x + \mu_n v) - f(x)}{\mu_n} \right) \right]$$

*is absolutely convergent and its limit is $\nabla_v f(x)$.*

*Proof.* First, because $\sum_{n=1}^{\infty} |a_n + b_n| \leqslant \sum_{n=1}^{\infty} |a_n| + \sum_{n=1}^{\infty} |b_n|$, it suffices to prove

$$\sum_{n=1}^{\infty} \left( \frac{f(x + \mu_{n+1} v) - f(x)}{\mu_{n+1}} - \frac{f(x + \mu_n v) - f(x)}{\mu_n} \right)$$

is absolutely convergent. To prove its absolute convergence, we estimate the magnitude of the difference term using Taylor's theorem:

$$\left| \frac{f(x + \mu_{n+1} v) - f(x)}{\mu_{n+1}} - \frac{f(x + \mu_n v) - f(x)}{\mu_n} \right|$$

$$\overset{(i)}{=} \left| \frac{\mu_{n+1} \nabla f(x)^\top v + R(x) \mu_{n+1}^2}{\mu_{n+1}} - \frac{\mu_n \nabla f(x)^\top v + R'(x) \mu_n^2}{\mu_n} \right|$$

$$\overset{(ii)}{\leqslant} \frac{L}{2} |\mu_{n+1} + \mu_n|$$

where (i) applies the Taylor theorem [Folland, 2002] with setting the integral form remainder $R(x) := \int_0^1 (1-t) \sum_{|\alpha|=2} \frac{\partial^2}{\partial x_1^{\alpha_1} \dots x_d^{\alpha_d}} f(x + t\mu v) \, dt$, (ii) assumes the global Lipschitz continuous gradient, which results in the uniform estimate $R(x) \leqslant \frac{L}{2}$ for all $x \in \mathbb{R}^d$. Therefore, to ensure the telescoping series is absolutely continuous, it suffices to require $\sum_{n=1}^{\infty} \mu_n < \infty$. The limit is directly determined by the original convergent series. It concludes our proof. $\square$

In the following two examples, we present commonly used sequences $\{\mu_n\}$ that satisfy the condition $\sum_{n=1}^{\infty} \mu_n < \infty$, ensuring the absolute convergence required in Proposition B.4.

**Example B.5** (Exponential Decay). Let $\mu_n = \alpha^n$ for some constant $0 < \alpha < 1$. Then,

$$\sum_{n=1}^{\infty} \mu_n = \sum_{n=1}^{\infty} \alpha^n = \frac{\alpha}{1 - \alpha} < \infty.$$

**Example B.6** (Polynomial Decay). Let $\mu_n = \frac{1}{n^s}$ for some constant $s > 1$. Then,

$$\sum_{n=1}^{\infty} \mu_n = \sum_{n=1}^{\infty} \frac{1}{n^s} < \infty,$$

which is well-known as the Riemann zeta function $\zeta(s)$.

## B.2 Unbiasedness of Zeroth-Order Estimators in $\mathscr{P}$-Family

We begin by recalling the definition of the unbiased zeroth-order gradient estimators, denoted by $\mathscr{P} := \mathscr{P}(f, \{\mu_n\}_{n=1}^{\infty}, \{p_n\}_{n=1}^{\infty}, V)$, as given by Definition 2.2. Under suitable conditions on the differentiable function $f : \mathbb{R}^d \to \mathbb{R}$ and the sequence $\{\mu_n\}_{n=1}^{\infty}$ (e.g., those provided by Proposition 2.1), this definition naturally yields the desired expectation representation. Moreover, the random distributions $\{p_n\}_{n=1}^{\infty}$ and $V$ are independent. These conditions are sufficient to guarantee the unbiasedness stated in the following result.

**Theorem B.7** (Unbiasedness). *Let $\mathscr{P} := \mathscr{P}(f, \{\mu_n\}_{n=1}^{\infty}, \{p_n\}_{n=1}^{\infty}, V)$ is defined as Definition 2.2. Then, for any estimator $\mathsf{P}(n, v) \in \mathscr{P}$, the following hold:*

(a) *$\mathbb{E}[\mathsf{P}(n, v) \mid v] = \nabla_v f(x)$; that is, $\mathsf{P}(n, v)$ is an unbiased estimator of the directional derivative $\nabla_v f(x)$.*

(b) *If the random direction $v$ is chosen independently of the sampling $n \sim \{p_n\}_{n=1}^{\infty}$ and satisfies $\mathbb{E}[v v^\top] = I$, then*

$$\mathbb{E}_{n \sim \{p_n\}_{n=1}^{\infty}, v \sim V}\left[\mathsf{P}(n, v) \, v\right] = \nabla f(x),$$

*so that $\mathsf{P}(n, v) \, v$ is an unbiased estimator of the full gradient.*

*Proof.* By Definition 2.2, the directional derivative $\nabla_v f(x)$ naturally has the expectation representation.

(a) Denote

$$X_n := \frac{f(x + \mu_1 v) - f(x)}{\mu_1} + \frac{1}{p_n}\left(\frac{f(x + \mu_{n+1} v) - f(x)}{\mu_{n+1}} - \frac{f(x + \mu_n v) - f(x)}{\mu_n}\right).$$

Then by the tower property of the conditional expectation,

$$\nabla_v f(x) = \mathbb{E}_{n \sim \{p_n\}_{n=1}^{\infty}}[X_n \mid v] = \mathbb{E}_{n \sim \{p_n\}_{n=1}^{\infty}}[\mathbb{E}[P(n, v)|n] \mid v] = \mathbb{E}[P(n, v) \mid v].$$

It concludes the proof.

(b) We consider the conditional expectation $\mathbb{E}[\cdot|v]$. We have

$$\begin{aligned}
\mathbb{E}\left[\mathsf{P}(n, v) \, v\right] &= \mathbb{E}\left[\mathbb{E}\left[\mathsf{P}(n, v) \, v \middle| v\right]\right] \\
&= \mathbb{E}\left[\mathbb{E}\left[\mathsf{P}(n, v) \middle| v\right] v\right] \\
&= \mathbb{E}[\nabla_v f(x) v] \\
&= \nabla f(x)
\end{aligned}$$

Therefore, we conclude that $\mathsf{P}(n, v) v$ is an unbiased estimator of the gradient $\nabla f(x)$.

$\square$

## C  Variance Analysis

In this section we present the proof of Theorem 3.1. Each subsection contains the proof of the corresponding item. We recap its statement below:

**Theorem C.1.** *Let $\mathscr{P} := \mathscr{P}(f, \{\mu_n\}_{n=1}^{\infty}, \{p_n\}_{n=1}^{\infty}, V)$ is defined as Definition 2.2. Suppose that $f : \mathbb{R}^d \to \mathbb{R}$ is second-order continuously differentiable and has L-Lipschitz continuous gradient, $\sum_{n=1}^{\infty} \mu_n < \infty$, and $V$ is the uniform distribution over the sphere with the radius $\sqrt{d}$. Define*

$$\mu := \mu_1, \qquad \varrho := \sum_{n=1}^{\infty} \frac{(\mu_{n+1} - \mu_n)^2}{p_n}, \qquad and \qquad \varphi := \sum_{n=1}^{\infty} \frac{\mu_n^2}{p_n}.$$

*Then the following statements hold:*

(a) *If there exists a point $x \in \mathbb{R}^d$ such that the Hessian $\nabla^2 f(x)$ is positive definite and $f(x) \neq 0$, then the variances of the $\mathsf{P}_1$ is infinite. That is,*

$$\mathrm{Var}[\mathsf{P}_1(n, v)v] = +\infty.$$

(b) *The variance of $\mathsf{P}_2$-estimator $\mathsf{P}_2(n, v)v$ is given by*

$$\mathrm{Var}[\mathsf{P}_2(n, v)\, v] \leqslant \mathrm{Var}[\mathsf{P}_4(n, v)\, v] + \frac{L^2}{3} d^3 \mu^2 + \frac{L^2}{12} d^3 \varrho + \frac{L^2}{3} d^3 \varphi.$$

(c) *The variance of $\mathsf{P}_3$-estimator $\mathsf{P}_3(n, v)v$ is given by*

$$\mathrm{Var}[\mathsf{P}_3(n, v)v] \leqslant \mathrm{Var}[\mathsf{P}_4(n, v)\, v] + \frac{L^2}{8} d^3 \mu^2 + \frac{L^2}{8} d^3 \varrho.$$

(d) *The variance of $\mathsf{P}_4$-estimator $\mathsf{P}_4(n, v)v$ is given by*

$$\mathrm{Var}[\mathsf{P}_4(n, v)v] \leqslant (d-1)\|\nabla f(x)\|^2 + \frac{3L^2}{4} d^3 \mu^2 + \frac{L^2 d^3}{2} \varrho.$$

*Proof.* For the item (a), we present the proof in Lemma C.2. For arbitrary $\mathsf{P} := \mathsf{P}(n, v) \in \mathscr{P}$, we have

$$\begin{aligned}
\mathrm{Var}[\mathsf{P}\, v] &\overset{(i)}{=} \mathbb{E}[\mathsf{P}^2 v^\top v] - \|\nabla f(x)\|^2 \\
&= d\mathbb{E}[(\mathsf{P} - \mathsf{P}_4(n, v) + \mathsf{P}_4(n, v))^2] - \|\nabla f(x)\|^2 \\
&\overset{(ii)}{=} d\mathbb{E}[(\mathsf{P} - \mathsf{P}_4(n, v))^2] + d\mathbb{E}[\mathsf{P}_4(n, v)] - \|\nabla f(x)\|^2 \\
&= d\mathbb{E}[(\mathsf{P} - \mathsf{P}_4(n, v))^2] + \mathrm{Var}[\mathsf{P}_4(n, v)\, v]
\end{aligned}$$

where (i) applies the unbiasedness (Theorem 2.3) and (ii) applies the definition of $\mathscr{P}$ ($\mathsf{P}$ is an unbiased estimator of $\mathsf{P}_4$). Therefore, we start with $\mathrm{Var}[\mathsf{P}_4\, v]$, the variance of the $\mathsf{P}_4$-estimator. Then it suffices to evaluate $\mathbb{E}[(\mathsf{P}_k(n, v) - \mathsf{P}_4(n, v))^2]$ for $k = 2$ and $k = 3$. Therefore, we prove the item (d) first. The detailed proof is included in Lemma C.3. Based on this result, we obtain the variance upper bound of $\mathsf{P}_2$- and $\mathsf{P}_3$-estimators in Lemma C.4 and Lemma C.5, respectively. $\qquad\square$

### C.1  Variance of $\mathsf{P}_1$-Estimator

**Lemma C.2.** *Under the same setting as Theorem C.1, if there exists a point $x \in \mathbb{R}^d$ such that the Hessian $\nabla^2 f(x)$ is positive definite, then the variances of the $\mathsf{P}_1$–estimator at $x$ is infinite.*

*Proof.* Recall that for the random direction $\frac{1}{\sqrt{d}} v \sim \mathrm{Uniform}(\mathbb{S}^{d-1})$ and the random variable $\mathsf{U}_4 \sim \mathrm{Uniform}(\{0, 1, 2, 3\})$, we have the $\mathsf{P}_1$-estimator defined as

$$\mathsf{P}_1(n, v) = \frac{f(x + \mu_1 v)\mathbb{I}_{\{\mathsf{U}_4 = 1\}} - f(x)\mathbb{I}_{\{\mathsf{U}_4 = 0\}}}{\mu_1}$$

$$+ \frac{1}{p_n} \left[ \frac{f(x + \mu_{n+1}v)\mathbb{I}_{\{U_4=2\}} - f(x)\mathbb{I}_{\{U_4=0\}}}{\mu_{n+1}} - \frac{f(x + \mu_n v)\mathbb{I}_{\{U_4=3\}} - f(x)\mathbb{I}_{\{U_4=0\}}}{\mu_n} \right].$$

Since this estimator is unbiased (Theorem 2.3),

$$\mathrm{Var}\big[\mathsf{P}_1(n,v)\,v\big] = d\mathbb{E}\big[\mathsf{P}_1(n,v)^2\big] - \|\nabla f(x)\|^2,$$

so it suffices to show $\mathbb{E}[\mathsf{P}_1(n,v)^2] = \infty$. For brevity we write

$$\mathsf{P}_1 := \mathsf{P}_1(n,v).$$

As the Hessian $\nabla^2 f(x)$ is positive definite at the point $x$, it has $\nabla^2 f(x) \succeq \lambda_{\min} I$ for some $\lambda_{\min} > 0$. We consider the event $\{N = n,\ U_3 = 3\}$, one finds

$$\mathsf{P}_1 = -\frac{f(x + \mu_n v)}{p_n \mu_n} \quad \text{with probability} \quad p_n \cdot \tfrac{1}{4}.$$

Hence

$$\mathbb{E}_{n \sim \{p_n\}}[\mathsf{P}_1^2] \geqslant \sum_{n=1}^{\infty} \mathrm{Pr}(N = n, U_3 = 3) \left( \frac{f(x + \mu_n v)}{p_n \mu_n} \right)^2$$

$$\geqslant \frac{1}{3} \sum_{n=1}^{\infty} \frac{[f(x + \mu_n v)]^2}{p_n \mu_n^2}$$

$$\geqslant \frac{1}{3} \sum_{n=1}^{\infty} \left( \frac{[f(x)]^2}{p_n \mu_n^2} + \frac{[f(x + \mu_n v) - f(x)]\, f(x)}{p_n \mu_n^2} \right)$$

Without loss of generality, we assume $f(x) > 0$. In the case $f(x) < 0$, we apply the $L$-smoothness to obtain a lower bound instead. By the second-order Taylor expansion [Spivak, 2008] (or the strong convexity near the point $x$),

$$f(x + \mu v) \geqslant f(x) + \mu \nabla f(x)^\top v + \tfrac{1}{2} \mu^2 \lambda_{\min} v^\top v.$$

so we have

$$\frac{[f(x + \mu_n v) - f(x)]\, f(x)}{p_n \mu_n^2} \geqslant \frac{\nabla f(x)^\top v}{p_n \mu_n} + \frac{d\lambda_{\min} f(x)}{2 p_n}.$$

As the result, we have

$$\mathbb{E}_{n \sim \{p_n\}, \frac{1}{\sqrt{d}} v \sim \mathrm{Uniform}(\mathbb{S}^{d-1})}[\mathsf{P}_1^2] \geqslant \frac{1}{3} \sum_{n=1}^{\infty} \frac{1}{p_n} \left( \frac{f(x)^2}{\mu_n^2} + \frac{d\lambda_{\min} f(x)}{2 p_n} \right).$$

As $\{p_n\}$ is a PMF, it must diverge to infinite. $\qquad\qquad\qquad\qquad\qquad\square$

## C.2 Variance of $\mathsf{P}_4$-Estimator

**Lemma C.3.** *Under the same setting as Theorem C.1, the variance of $\mathsf{P}_4$-estimator $\mathsf{P}_4(n,v)\,v$ is upper bounded by*

$$\mathrm{Var}[\mathsf{P}_4(n,v)\,v] \leqslant (d-1)\|\nabla f(x)\|^2 + \frac{3L^2 d^3 \mu_1^2}{4} + \frac{L^2 d^3}{2} \sum_{n=1}^{\infty} \frac{|\mu_{n+1} - \mu_n|^2}{p_n}.$$

*Proof.* Recall that

$$\mathsf{P}_4(n,v) = \frac{f(x + \mu_1 v) - f(x)}{\mu_1} + \frac{1}{p_n}\left[ \frac{f(x + \mu_{n+1}v) - f(x)}{\mu_{n+1}} - \frac{f(x + \mu_n v) - f(x)}{\mu_n} \right].$$

Our goal is to bound $\mathrm{Var}[\mathsf{P}_4(n,v)\,v]$. For each $n$, define the remainder term

$$\delta_n(v) = \frac{f(x + \mu_n v) - f(x) - \mu_n \nabla f(x)^\top v}{\mu_n},$$

$$\Delta_n = \frac{f(x + \mu_n v) - f(x)}{\mu_n} = \delta_n(v) + \nabla f(x)^\top v.$$

Then
$$P_4(n, v) = \nabla f(x)^\top v + \delta_1(v) + \frac{\delta_{n+1}(v) - \delta_n(v)}{p_n}.$$

Hence our vector estimator is
$$P_4(n, v)\, v = vv^\top \nabla f(x) + \Big[\delta_1(v) + \tfrac{\delta_{n+1}(v) - \delta_n(v)}{p_n}\Big] v.$$

Since we have shown that $P_4(n, v)\, v$ is unbiased (Theorem 2.3) and $\mathbb{E}[vv^\top] = I_d$, we have
$$\mathbb{E}\Big[\delta_1(v) + \tfrac{\delta_{n+1}(v) - \delta_n(v)}{p_n}\Big] v = 0. \tag{11}$$

The variance is given by
$$\begin{aligned}
\mathrm{Var}[P_4(n, v)\, v] &= d\mathbb{E}[P_4(n, v)^2] - \|\nabla f(x)\|^2 \\
&= (d-1)\|\nabla f(x)\|^2 + d\mathbb{E}\left(\delta_1(v) + \frac{\delta_{n+1}(v) - \delta_n(v)}{p_n}\right)^2 \\
&\quad + 2d\mathbb{E}\nabla f(x)^\top v\left(\delta_1(v) + \frac{\delta_{n+1}(v) - \delta_n(v)}{p_n}\right) \\
&\overset{(i)}{=} (d-1)\|\nabla f(x)\|^2 + d\mathbb{E}\left(\delta_1(v) + \frac{\delta_{n+1}(v) - \delta_n(v)}{p_n}\right)^2 \\
&= (d-1)\|\nabla f(x)\|^2 + d\mathbb{E}\delta_1^2(v) \\
&\quad + 2d\mathbb{E}\left[\frac{\delta_1(v)}{p_n}\left(\delta_{n+1}(v) - \delta_n(v)\right)\right] + \mathbb{E}\left[\frac{d}{p_n^2}\left(\delta_{n+1}(v) - \delta_n(v)\right)^2\right] \\
&\overset{(ii)}{=} (d-1)\|\nabla f(x)\|^2 + 3d\mathbb{E}\delta_1^2(v) + \mathbb{E}\left[\frac{d}{p_n^2}\left(\delta_{n+1}(v) - \delta_n(v)\right)^2\right]
\end{aligned}$$

where (i) applies the unbiasedness Eq. (11) and (ii) applies the telescoping series Eq. (2). Now it remains to upper bound the remainder term $\delta_n(v)$ and the remainder difference term $\delta_{n+1}(v) - \delta_n(v)$.

- **Bound the remainder term $\delta_n(v)$:** By $L$-smoothness of $f : \mathbb{R}^d \to \mathbb{R}$, we have

$$f(x + \mu_n v) - f(x) \leqslant \mu_n \nabla f(x)^\top v + \frac{L\mu_n^2}{2}\|v\|^2,$$
$$\overset{(i)}{=} \mu_n \nabla f(x)^\top v + \frac{dL\mu_n^2}{2},$$

  where (i) applies $\mathbb{E}vv^\top = I_d$ (this condition ensures that $\|v\|^2 = \mathrm{Tr}\left(\|v\|^2\right) = \mathrm{Tr}\left(vv^\top\right) = d$). As the result,

$$\delta_n(v) \leqslant \frac{Ld}{2}\mu_n. \tag{12}$$

  or we may use the following almost-sure upper bound

$$[\delta_n(v)]^2 \leqslant \frac{L^2 d^2}{4}\mu_n^2.$$

- **Bound the remainder difference term $\delta_{n+1}(v) - \delta_n(v)$:** We define the remainder term as a function in $\mu$; that is,

$$\phi(\mu) := \frac{f(x + \mu v) - f(x) - \mu \nabla f(x)^\top v}{\mu}.$$

  It automatically gives

$$\delta_{n+1}(v) - \delta_n(v) = \phi(\mu_{n+1}) - \phi(\mu_n).$$

As $\phi : [\mu_{n+1}, \mu_n] \to \mathbb{R}$ is a continuous differentiable function (by our assumption that $f : \mathbb{R}^d \to \mathbb{R}$ is second-order continuously differentiable), we can apply the mean-value theorem [Folland, 2002]: There exists $\varsigma \in [\mu_{n+1}, \mu_n]$ such that

$$\phi(\mu_{n+1}) - \phi(\mu_n) = \phi'(\varsigma)(\mu_{n+1} - \mu_n).$$

We again applying the $L$-smoothness of $f : \mathbb{R}^d \to \mathbb{R}$ (essentially the bounded Hessian assumption) to $\phi'(\varsigma)$, which leads to

$$|\delta_{n+1}(v) - \delta_n(v)| \leqslant \frac{Ld}{2}|\mu_{n+1} - \mu_n|. \tag{13}$$

As the result, we obtain the upper bound as

$$\mathrm{Var}[\mathsf{P}_4(n,v)\,v] \overset{(i)}{\leqslant} (d-1)\|\nabla f(x)\|^2 + \frac{3L^2 d^3 \mu_1^2}{4} + \sum_{n=1}^{\infty} \frac{2d}{p_n}\left(\frac{Ld}{2}|\mu_{n+1} - \mu_n|\right)^2$$

$$\leqslant (d-1)\|\nabla f(x)\|^2 + \frac{3L^2 d^3 \mu_1^2}{4} + \frac{L^2 d^3}{2}\sum_{n=1}^{\infty}\frac{|\mu_{n+1} - \mu_n|^2}{p_n},$$

where (i) applies the expectation over $n \sim \{p_n\}_{n=1}^{\infty}$, which cancels out one $\frac{1}{p_n}$. $\qquad\square$

## C.3 Variance of $\mathsf{P}_2$-Estimator

**Lemma C.4.** *Under the same setting as Theorem C.1, the variance of $\mathsf{P}_2$-estimator $\mathsf{P}_2(n,v)\,v$ is upper bounded by*

$$\mathrm{Var}[\mathsf{P}_2(n,v)\,v] \leqslant \mathrm{Var}[\mathsf{P}_4(n,v)\,v] + \frac{L^2}{12}d^3\sum_{n=1}^{\infty}\frac{(\mu_n - \mu_{n+1})^2}{p_n} + \frac{L^2}{3}d^3\mu_1^2 + \frac{L^2}{3}d^3\sum_{n=1}^{\infty}\frac{\mu_n^2}{p_n}.$$

*Proof.* Recall that $\mathsf{P}_2(n,v)$ is defined as

$$\mathsf{P}_2(n,v) = \frac{f(x + \mu_1 v) - f(x)}{\mu_1}\mathbb{I}_{\{U_3 = 0\}}$$

$$+ \frac{1}{p_n}\left[\frac{f(x + \mu_{n+1}v) - f(x)}{\mu_{n+1}}\mathbb{I}_{\{U_3 = 1\}} - \frac{f(x + \mu_n v) - f(x)}{\mu_n}\mathbb{I}_{\{U_3 = 2\}}\right],$$

where $U_3 \sim \mathrm{Uniform}(\{0,1,2\})$ is a selection variable. Then

$$\mathbb{E}[(\mathsf{P}_2(n,v) - \mathsf{P}_4(n,v))^2 \mid n, v]$$

$$= \mathbb{P}(U_3 = 0)\left[\frac{1}{p_n}\left[\frac{f(x + \mu_{n+1}v) - f(x)}{\mu_{n+1}} - \frac{f(x + \mu_n v) - f(x)}{\mu_n}\right]\right]^2$$

$$+ \mathbb{P}(U_3 = 1)\left[\frac{f(x + \mu_1 v) - f(x)}{\mu_1} + \frac{1}{p_n}\left[-\frac{f(x + \mu_n v) - f(x)}{\mu_n}\right]\right]^2$$

$$+ \mathbb{P}(U_3 = 2)\left[\frac{f(x + \mu_1 v) - f(x)}{\mu_1} + \frac{1}{p_n}\left[\frac{f(x + \mu_{n+1}v) - f(x)}{\mu_{n+1}}\right]\right]^2$$

$$\overset{(i)}{\leqslant} \frac{1}{3}\frac{1}{p_n^2}(\delta_{n+1}(v) - \delta_n(v))^2 + \frac{2}{3}\left([\delta_1(v)]^2 + \frac{1}{p_n^2}[\delta_n(v)]^2\right) + \frac{2}{3}\left([\delta_1(v)]^2 + \frac{1}{p_n^2}[\delta_{n+1}(v)]^2\right)$$

$$= \frac{4}{3}[\delta_1(v)]^2 + \frac{1}{3}\frac{1}{p_n^2}(\delta_{n+1}(v) - \delta_n(v))^2 + \frac{2}{3}\frac{1}{p_n^2}[\delta_n(v)]^2 + \frac{2}{3}\frac{1}{p_n^2}[\delta_{n+1}(v)]^2,$$

where (i) applies $(a+b)^2 \leqslant 2a^2 + 2b^2$ and $\delta_n(v) := \frac{f(x + \mu_n v) - f(x) - \mu_n \nabla f(x)^\top v}{\mu_n}$. As we have bounded this term in Lemma C.3, we have

$$\mathbb{E}[(\mathsf{P}_2(n,v) - \mathsf{P}_4(n,v))^2 \mid v]$$

$$= \sum_{n=1}^{\infty} p_n [(\mathsf{P}_2(n,v) - \mathsf{P}_4(n,v))^2 \mid n, v]$$

$$\leqslant \sum_{n=1}^{\infty} p_n \left[ \frac{L^2 d^2}{3} \mu_1^2 + \frac{L^2 d^2}{12} \frac{(\mu_n - \mu_{n+1})^2}{p_n^2} + \frac{L^2 d^2}{3 p_n^2} \mu_n^2 \right]$$

$$\leqslant \frac{L^2}{12} d^2 \sum_{n=1}^{\infty} \frac{(\mu_n - \mu_{n+1})^2}{p_n} + \frac{L^2}{3} d^2 \mu_1^2 + \frac{L^2}{3} d^2 \sum_{n=1}^{\infty} \frac{\mu_n^2}{p_n}.$$

Therefore, we finally obtain

$$\mathrm{Var}[\mathsf{P}_2(n,v)\, v] = d\mathbb{E}[(\mathsf{P}_2(n,v) - \mathsf{P}_4(n,v))^2] + \mathrm{Var}[\mathsf{P}_4(n,v)\, v]$$

$$\leqslant \mathrm{Var}[\mathsf{P}_4(n,v)\, v] + \frac{L^2}{12} d^3 \sum_{n=1}^{\infty} \frac{(\mu_n - \mu_{n+1})^2}{p_n} + \frac{L^2}{3} d^3 \mu_1^2 + \frac{L^2}{3} d^3 \sum_{n=1}^{\infty} \frac{\mu_n^2}{p_n}.$$

It completes the proof. □

### C.4  Variance of $\mathsf{P}_3$-Estimator

**Lemma C.5.** *Under the same setting as [Theorem C.1](#), the variance of $\mathsf{P}_3$-estimator $\mathsf{P}_3(n,v)\, v$ is upper bounded by*

$$\mathrm{Var}[\mathsf{P}_3(n,v)\, v] \leqslant \mathrm{Var}[\mathsf{P}_4(n,v)\, v] + \frac{L^2}{8} d^3 \sum_{n=1}^{\infty} \frac{(\mu_n - \mu_{n+1})^2}{p_n} + \frac{L^2}{8} d^3 \mu_1^2.$$

*Proof.* Recall that $\mathsf{P}_3(n,v)$ is defined as

$$\mathsf{P}_3(n,v)$$
$$= \frac{f(x + \mu_1 v) - f(x)}{\mu_1} U_2 + \frac{1}{p_n} \left[ \frac{f(x + \mu_{n+1} v) - f(x)}{\mu_{n+1}} - \frac{f(x + \mu_n v) - f(x)}{\mu_n} \right] (1 - U_2),$$

where $U_2 \sim \mathrm{Uniform}\,(\{0,1\})$ is a selection variable. Then

$$\mathbb{E}[(\mathsf{P}_3(n,v) - \mathsf{P}_4(n,v))^2 \mid n, v]$$

$$= \mathbb{P}(U_2 = 0) \left[ \frac{1}{p_n} \left[ \frac{f(x + \mu_{n+1} v) - f(x)}{\mu_{n+1}} - \frac{f(x + \mu_n v) - f(x)}{\mu_n} \right] - \mathsf{P}_4(n,v) \right]^2$$

$$+ \mathbb{P}(U_2 = 1) \left[ \frac{f(x + \mu_1 v) - f(x)}{\mu_1} - \mathsf{P}_4(n,v) \right]^2$$

$$= \frac{1}{2} \left[ \frac{1}{p_n} \left[ \frac{f(x + \mu_{n+1} v) - f(x)}{\mu_{n+1}} - \frac{f(x + \mu_n v) - f(x)}{\mu_n} \right] \right]^2 + \frac{1}{2} \left[ \frac{f(x + \mu_1 v) - f(x)}{\mu_1} \right]^2.$$

As we have bounded this term in [Lemma C.3](#), we have

$$\mathbb{E}[(\mathsf{P}_3(n,v) - \mathsf{P}_4(n,v))^2 \mid v] = \sum_{n=1}^{\infty} p_n [(\mathsf{P}_3(n,v) - \mathsf{P}_4(n,v))^2 \mid n, v]$$

$$\leqslant \sum_{n=1}^{\infty} p_n \left[ \frac{1}{p_n^2} \frac{L^2 d^2}{8} |\mu_{n+1} - \mu_n|^2 + \frac{L^2 d^2 \mu_1^2}{8} \right]$$

$$\leqslant \frac{L^2}{8} d^2 \sum_{n=1}^{\infty} \frac{(\mu_n - \mu_{n+1})^2}{p_n} + \frac{L^2}{8} d^2 \mu_1^2.$$

Therefore, we finally obtain

$$\mathrm{Var}[\mathsf{P}_3(n,v)\, v] = d\mathbb{E}[(\mathsf{P}_3(n,v) - \mathsf{P}_4(n,v))^2] + \mathrm{Var}[\mathsf{P}_4(n,v)\, v]$$

$$\leqslant \mathrm{Var}[\mathsf{P}_4(n,v)\, v] + \frac{L^2}{8} d^3 \sum_{n=1}^{\infty} \frac{(\mu_n - \mu_{n+1})^2}{p_n} + \frac{L^2}{8} d^3 \mu_1^2.$$

It concludes the proof. □

## C.5 On the Optimal Sampling Distribution and the Perturbation Stepsize Sequence

We recap the full statement of Theorem 3.2.

**Theorem C.6.** *Let $\{\mu_n\}_{n=1}^{\infty}$ be a positive, decreasing sequence with $\sum_{n=1}^{\infty} \mu_n < \infty$, and let $\{p_n\}_{n=1}^{\infty}$ be a PMF. Then the following statements hold:*

*(a) Define $\varrho_n = \frac{(\mu_{n+1}-\mu_n)^2}{p_n}$. The lower bound of $\varrho$ is given by*

$$\varrho = \sum_{n=1}^{\infty} \varrho_n \geqslant \mu_1^2.$$

*Moreover, equality holds if and only if $p_n = \frac{\mu_n - \mu_{n+1}}{\mu_1}$.*

*(b) Define $\varphi_n = \frac{\mu_n^2}{p_n}$. The lower bound of $\varphi$ is given by*

$$\varphi = \sum_{n=1}^{\infty} \varphi_n \geqslant \Big( \sum_{n=1}^{\infty} \mu_n \Big)^2 > \mu_1^2.$$

*Moreover, equality holds if and only if $p_n = \frac{\mu_n}{\sum_{n=1}^{\infty} \mu_n}$.*

*Proof.* Write $a_n = \mu_{n+1} - \mu_n$, so $\sum_{n=1}^{\infty} a_n = \mu_1$ and $\sum_n p_n = 1$. By Cauchy-Schwarz inequality,

$$\Big( \sum_n |a_n| \Big)^2 \leqslant \Big( \sum_n p_n \Big) \Big( \sum_n \frac{a_n^2}{p_n} \Big) = \sum_{n=1}^{\infty} \varrho_n,$$

which yields the claimed lower bound in (a). Equality occurs exactly when $p_n \propto |a_n|$, i.e. $p_n = (\mu_n - \mu_{n+1})/\mu_1$. Similarly, By Cauchy-Schwarz inequality,

$$\Big( \sum_n \mu_n \Big)^2 \leqslant \Big( \sum_n \frac{\mu_n^2}{p_n} \Big) \Big( \sum_n p_n \Big) = \sum_{n=1}^{\infty} \varphi_n,$$

which yields the claimed lower bound in (b). $\qquad \square$

In the following example, we include the omitted details of Example 3.4.

**Example C.7.** We consider the Zipf distribution $n \sim \mathrm{Zipf}(s)$ $(s > 1)$. Then

$$p_n = \frac{1}{\zeta(s)} \frac{1}{n^s}, \qquad \zeta(s) = \sum_{n=1}^{\infty} \frac{1}{n^s}.$$

We define $\{\mu_n\}$ by the recursion

$$\mu_n - \mu_{n+1} = \mu_1 \, p_n = \mu_1 \frac{1}{\zeta(s)} \frac{1}{n^s},$$

so that summing gives the closed-form

$$\mu_n = \mu_1 \Big( 1 - \frac{\sum_{j=1}^{n-1} j^{-s}}{\zeta(s)} \Big) = \mu_1 \frac{\sum_{j=n}^{\infty} j^{-s}}{\zeta(s)}.$$

A direct check shows this choice attains the lower bound $\varrho = \mu_1^2$ on $\sum (\mu_n - \mu_{n+1})^2 / p_n$. Now we turn to bound $\varphi$:

$$\varphi = \sum_{n=1}^{\infty} \frac{\mu_n^2}{p_n} = \mu_1^2 \, \zeta(s) \sum_{n=1}^{\infty} n^s \Big( 1 - \frac{\sum_{j=1}^{n-1} j^{-s}}{\zeta(s)} \Big)^2 = \mu_1^2 \, \zeta(s) \sum_{n=1}^{\infty} n^s \Big( \frac{\sum_{j=n}^{\infty} j^{-s}}{\zeta(s)} \Big)^2.$$

For $s > 3$, use the integral bound

$$\sum_{j=n}^{\infty} j^{-s} \leqslant \int_{n-1}^{\infty} x^{-s} \, dx = \frac{(n-1)^{1-s}}{s-1},$$

to get

$$\frac{\mu_n^2}{p_n} \leqslant \frac{\mu_1^2}{(s-1)^2 \zeta(s)} n^{2-s}.$$

Since $s > 3$ the series $\sum_{n=1}^{\infty} n^{2-s} = \zeta(s-2)$ converges, giving the clean bound

$$\varphi = \sum_{n=1}^{\infty} \frac{\mu_n^2}{p_n} \leqslant \frac{\zeta(s-2)}{(s-1)^2 \zeta(s)} \mu_1^2.$$

## C.6 Discussions: Variance of Random Directional Derivative

In this subsection, we analyze the variance of a gradient estimate based on a random directional derivative. Let $v$ be a random vector uniformly sampled from the sphere with the dimension $d$. We approximate the gradient $\nabla f(x)$ using the random directional derivative defined as

$$\nabla_v f(x) := vv^\top \nabla f(x).$$

Assuming that the expectation satisfies $\mathbb{E}[vv^\top] = I$, it is essential to evaluate the variance of this estimator. Specifically, we compute:

$$\mathbb{E}[\nabla f(x)^\top vv^\top vv^\top \nabla f(x)] = \mathbb{E}[\|\nabla f(x)\|^2 \|v\|^2]$$
$$= d\|\nabla f(x)\|^2.$$

Thus, the variance is given by

$$\mathrm{Var}[\nabla_v f(x)] = d\|\nabla f(x)\|^2 - \|\nabla f(x)\|^2 = (d-1)\|\nabla f(x)\|^2.$$

This result indicates that even when the exact directional derivative is available, the variance still scales with $\mathcal{O}(d)$ relative to the gradient norm. Consequently, it is not avoidable to remove the dependence on the dimension $d$.

## D    Convergence Analysis

In this section, we present the proof of Corollary 3.5. Recall that our goal is to solve the stochastic optimization problem Eq. (1):

$$\min_{x \in \mathbb{R}^d} f(x) := \mathbb{E}_{\xi \sim \Xi} f(x; \xi),$$

where the second-order continuously differentiable function $f(\cdot; \xi) : \mathbb{R}^d \to \mathbb{R}$ has $L$-Lipschitz gradient for every $\xi$. We consider the convergence upper bound of the classical stochastic gradient descent (SGD) algorithm with the constant learning rate $\eta$ given the initialization $x_0$:

$$x_{t+1} = x_t - \eta g(x_t), \qquad \text{(SGD)}$$

where $g(x_t)$ is an unbiased estimator of the full gradient $\nabla f(x)$.

### D.1    The Convergence Upper Bound

The full statement of Corollary 3.5 is given as follow:

**Corollary D.1.** *Consider the stochastic optimization problem in Eq. (1), and suppose that the individual loss $f(x; \xi)$ is second-order differentiable with L-Lipschitz continuous gradient in $x$, uniformly over $\xi \sim \Xi$. Assume the stochastic gradient is approximated using the $\mathsf{P}_k$-estimator $\mathsf{P}_k(n, v) \, v$ for $k = 2, 3, 4$. Let the SGD iteration be defined as*

$$x_{t+1} = x_t - \eta \mathsf{P}_k(n_t, v_t) \, v_t$$

*where $\eta \in (0, \frac{1}{L^2 d}]$ is the stepsize. Then the iterates satisfy the following convergence guarantee:*

$$\min_{0 \leqslant t \leqslant T-1} \mathbb{E}\|\nabla f(x_t)\|^2 \leqslant L\eta \left[ C_k d^3 \mu^2 + 2dL(f^* - \mathbb{E}_{\xi \sim \Xi} f_\xi^*) \right] + \frac{2}{\eta T} \delta,$$

where $\underline{\delta} := f(x_0) - f^*$ and $C_k = \begin{cases} \frac{28L^2}{12} & k = 2, \\ \frac{3L^2}{4} & k = 3, \\ \frac{5L^2}{4} & k = 4, \end{cases}$ $(k = 2, 3, 4)$ is the estimator-dependent error term. Consequently, choosing $\eta = \Theta(1/\sqrt{dT})$ and $\mu = \mathcal{O}(\frac{1}{d})$ yields the optimal complexity $T = \Theta(\frac{d}{\epsilon^4})$ of having $\min_{0 \leqslant t \leqslant T-1} \mathbb{E}\|\nabla f(x_t)\| \leqslant \epsilon$.

*Proof.* Let $g(x) = \mathsf{P}_k(n, v)\, v$ for $k = 2, 3, 4$. By Lemma D.4, we have

$$\mathbb{E}\|g(x)\| \leqslant dL\|\nabla f(x)\|^2 + C_k d^3 \mu^2 + 2dL(f^* - \mathbb{E}_{\xi \sim \Xi} f_\xi^*).$$

Then we set $B = dL$ and $C = C_k d^3 \mu^2 + 2dL(f^* - \mathbb{E}_{\xi \sim \Xi} f_\xi^*)$ in Lemma D.3. As the result, when $\eta \leqslant \frac{1}{L^2 d}$, we have

$$\min_{0 \leqslant t \leqslant T-1} \mathbb{E}\|\nabla f(x_t)\|^2 \leqslant L\eta \left[ C_k d^3 \mu^2 + 2dL(f^* - \mathbb{E}_{\xi \sim \Xi} f_\xi^*) \right] + \frac{2}{\eta T}\underline{\delta}.$$

The complexity of making $\min_{0 \leqslant t \leqslant T-1} \mathbb{E}\|\nabla f(x_t)\| \leqslant \epsilon$ is given by

$$T \geqslant \frac{12\underline{\delta} L}{\epsilon^2} \max \left\{ dL, 2\frac{C_k d^3 \mu^2 + 2dL(f^* - \mathbb{E}_{\xi \sim \Xi} f_\xi^*)}{\epsilon^2} \right\}.$$

Setting $\mu = \mathcal{O}(\frac{1}{d})$, the optimal complexity is given by $T = \Theta(\frac{d}{\epsilon^4})$. $\qquad\square$

### D.2 Supporting Lemmas

**Lemma D.2.** *Let $f^* := \min_x f(x)$ and $f_\xi^* := \min_x f(x; \xi)$. If for each $\xi$, $f(x; \xi)$ has L-Lipschitz gradient, then*
$$\mathbb{E}\|\nabla f(x; \xi)\|^2 \leqslant L\|\nabla f(x)\|^2 + 2L(f^* - \mathbb{E}_{\xi \sim \Xi} f_\xi^*).$$

*Proof.* This lemma is directly taken from Proposition 2, Khaled and Richtárik [2022]. $\qquad\square$

**Lemma D.3.** *Let the second-order continuously differentiable function $f(\cdot\, ; \xi) : \mathbb{R}^d \to \mathbb{R}$ be lower bounded by $f^* := \min_x f(x)$ and have L-Lipschitz gradient for every $\xi$, and $g(x)$ be an unbiased estimator of $\nabla f(x)$. Suppose that $g(x)$ and $f(x)$ satisfy the expected smoothness condition*

$$\mathbb{E}\|g(x)\|^2 \leqslant B \cdot \|\nabla f(x)\|^2 + C.$$

*If the learning rate $\eta \leqslant \frac{1}{LB}$ and define $\underline{\delta} := f(x_0) - f^*$, then the $T$-th iteration of SGD satisfies*

$$\min_{0 \leqslant t \leqslant T-1} \mathbb{E}\|\nabla f(x_t)\|^2 \leqslant LC\eta + \frac{2}{\eta T}\underline{\delta}.$$

*Proof.* This lemma is directly taken from Theorem 2, Khaled and Richtárik [2022]. $\qquad\square$

**Lemma D.4.** *Let $g(x)$ be the $\mathsf{P}_k$-estimator (for $k = 2, 3, 4$), $f^* := \min_x f(x)$, and $f_\xi^* := \min_x f(x; \xi)$. Then the expected smoothness condition is given by*

$$\mathbb{E}\|g(x)\| \leqslant dL\|\nabla f(x)\|^2 + C_k d^3 \mu^2 + 2dL(f^* - \mathbb{E}_{\xi \sim \Xi} f_\xi^*).$$

*where*

$$C_k = \begin{cases} \frac{28L^2}{12} & k = 2, \\ \frac{3L^2}{4} & k = 3, \\ \frac{5L^2}{4} & k = 4, \end{cases}$$

*is the error term introduced by the zeroth-order gradient estimation. For $\mathsf{P}_2$-estimator, we choose $\{\mu_n\}$ and $\{p_n\}$ such that $\varrho = \mu^2$ and $\varphi \leqslant 2\mu^2$.*

*Proof.* Let $\hat{\nabla} f(x;\xi)$ be the $\mathsf{P}_k$-estimator for $k = 2, 3, 4$. First, we notice the following variance-decomposition holds:

$$\mathbb{E}\|\hat{\nabla} f(x;\xi)\|^2 = \mathbb{E}\|\hat{\nabla} f(x;\xi) - \nabla f(x;\xi) + \nabla f(x;\xi)\|^2$$
$$= \mathbb{E}\|\nabla f(x;\xi)\|^2 + \mathbb{E}\left[\text{Var}\left[\mathsf{P}_k\, v \mid \xi\right]\right].$$

By Theorem 3.1 and Theorem 3.2, we set

$$\text{Var}\left[\mathsf{P}_k\, v \mid \xi\right] \leqslant (d-1)\|\nabla f(x;\xi)\|^2 + C_k d^3 \mu^2,$$

where $C_k$ is determined by the estimator; we also assume that the optimal distribution and perturbations are taken obeying Theorem 3.2. As the result,

$$\mathbb{E}\|\hat{\nabla} f(x;\xi)\|^2 \leqslant \mathbb{E}\|\nabla f(x;\xi)\|^2 + +\mathbb{E}\left[(d-1)\|\nabla f(x;\xi)\|^2 + C_k d^3 \mu^2\right]$$
$$\leqslant d\mathbb{E}\|\nabla f(x;\xi)\|^2 + C_k d^3 \mu^2$$
$$\overset{(i)}{\leqslant} dL\|\nabla f(x)\|^2 + C_k d^3 \mu^2 + 2dL(f^* - \mathbb{E}_{\xi\sim\Xi} f_\xi^*),$$

where (i) applies Lemma D.2. It completes the proof. $\square$

# E  Experiments Details

In this section, we describe the detailed experiment setting and the hyperparameter configurations.

## E.1  Synthetic Example

In the synthetic example, we compared gradient estimators across varying dimensions ($d = 16, 64, 256, 1024, 4096$) using both quadratic and logistic functions. For fair comparison, all estimators used a consistent number of function evaluations of 3 and the perturbation stepsize $\mu = 10^{-5}$ (for $\mathsf{P}_3$-estimator, we set $\mu_1 = \mu$). The $\mathsf{P}_3$-estimator was configured with parameter $s = 2.0$ and followed the same perturbation stepsize scheduling as Example 3.4. We evaluated four gradient estimators: Zipf's $P_3$-estimator, one-side two-point estimator with the Gaussian smoothing, one-side two-point estimator with the uniform smoothing, and two-side two-point estimator with the Gaussian smoothing. Each configuration was tested over 100 independent trials with a fixed random seed for reproducibility.

**Code Availability and System Requirements**   All source code is included in the supplementary materials. No specific hardware is required; any machine supporting Python 3.10.10 should suffice. A Jupyter notebook version is also provided for convenient execution on Google Colab.

## E.2  Language Model Optimization

In the language model optimization experiment, we compare the performance of different gradient estimators within a vanilla SGD framework for fine-tuning a language model on a sentiment classification task. The learning rate of SGD is fixed at $\eta = 10^{-4}$, the batch size of SGD is fixed at 16 (this batch size corresponds to the number of stochastic samples and is different from the batch size in multiple-point zeroth-order estimator), and the perturbation stepsize is set to $\mu = 10^{-3}$ (for our proposed unbiased estimators, we use $\mu_1 = \mu$). Due to limitations in numerical precision, we do not sample the extreme tail of the distribution. Instead, we truncate the sampling distribution by enforcing $p_n \geqslant 10^{-3}$ to ensure numerical stability and avoid excessively small probabilities. All remaining hyperparameters are summarized in Table 1.

**Code Availability and System Requirements**   All source code and reproduction instructions are provided in the supplementary materials. Experiments were conducted on a cluster running RHEL 8, equipped with dual AMD EPYC 7763 processors, 512 GB of memory, and seven NVIDIA RTX 5000 GPUs. To reproduce the language model optimization experiments, we recommend using a CUDA-compatible GPU with at least 24 GB of VRAM.

Table 1: Summary of gradient estimators used in the language model optimization experiment.

| Estimation Method | Estimator Formula | Batch Size $b$ | # Function Calls | Notes |
|---|---|---|---|---|
| One-Side Two-Point Estimator | $\sum_{i=1}^{b} \dfrac{f(x + \mu v_i) - f(x)}{\mu} v_i$ | 3 | 4 | $v_i \overset{iid}{\sim} \mathrm{Normal}(0, I_d)$ |
| Two-Side Two-Point Estimator | $\sum_{i=1}^{b} \dfrac{f(x + \mu v_i) - f(x - \mu v_i)}{2\mu} v_i$ | 2 | 4 | $v_i \overset{iid}{\sim} \mathrm{Normal}(0, I_d)$ |
| Zipf's $P_3$-Estimator | Eq. (7) | 1 | 3 | $s = 1.5$, Example 3.4 |
| Geometric $P_3$-Estimator | Eq. (7) | 1 | 3 | $c = 0.5$, Example 3.3 |
| Zipf's $P_4$-Estimator | Eq. (6) | 1 | 4 | $s = 1.5$, Example 3.4 |
| Geometric $P_4$-Estimator | Eq. (6) | 1 | 4 | $c = 0.5$, Example 3.3 |

# F   Broader Impact

This work introduces a new class of unbiased zeroth-order gradient estimators that offer both theoretical guarantees and practical advantages. By eliminating bias without increasing variance, our method enhances the reliability of optimization in settings where gradient information is unavailable or costly to obtain. These include fine-tuning large language models under memory constraints, conducting black-box adversarial robustness evaluations, and solving scientific computing tasks such as physics-informed neural networks. On the theoretical side, our estimators advance the understanding of zeroth-order optimization and provide new tools for the zeroth-order gradient estimation. This work opens a promising direction for future research in gradient-free optimization and its broad applications in machine learning and beyond.

# G   Limitations

Despite the theoretical guarantees and empirical improvements demonstrated by our proposed unbiased zeroth-order gradient estimators, several limitations remain. First, the estimators rely on sampling from an infinite sequence of perturbation steps, but practical implementations must truncate this sequence, which may reintroduce bias or affect variance control. Second, the proposed estimators are based on directional derivatives, which inherently exhibit a dependence on the problem dimension $d$; this dimensional dependence is generally unavoidable for this class of methods. Lastly, while we validate the approach on synthetic tasks and language model fine-tuning, we have not extensively evaluated its performance across a broader range of optimization problems, and the observed empirical gains may not fully generalize to settings involving non-smooth objectives or high levels of evaluation noise.

