# OpenReview forum: "On the Optimal Construction of Unbiased Gradient Estimators for Zeroth-Order Optimization"
_NeurIPS.cc/2025/Conference — NeurIPS 2025 spotlight_

### Official Review · Reviewer_Yvd5 · 2025-06-13

**Clarity:** 2
**Significance:** 3
**Originality:** 3
**Rating:** 5
**Confidence:** 4

**Summary:**

This paper introduces a framework for constructing unbiased estimators of the true directional derivative of a smooth function using only zeroth-order queries. The authors reformulate the directional derivative as an infinite series and, leveraging the absolute convergence of this series, derive an expectation-based formulation. Within this framework, they propose and analyze specific estimators, providing theoretical guarantees on the expected stationarity gap in the context of non-convex stochastic optimization.

**Questions:**

1. The result that unbiased estimators can attain the same level of variance in the smooth setting as standard single- or two-point estimators is certainly interesting. However, it remains unclear how this observation contributes either theoretically or practically. Specifically, does exact unbiasedness offer any concrete advantages over the existing analyses of zeroth-order optimization that rely on the gradient of a smoothed function?

2. The construction of these estimators involves selecting a countably infinite set of distributions $p_n$ and corresponding smoothing parameters $\mu_n$. It is unclear how this construction guarantees exact unbiasedness in practice. Could this be interpreted instead as an approximation or a limiting form of unbiasedness? If so, why would this approach be preferable to, or more suitable than, using standard smoothed gradient estimators in algorithmic applications?

**Ethical Concerns:**

["NO or VERY MINOR ethics concerns only"]

**Final Justification:**

My initial concern was centered on the novelty and significance of the work. I had mistakenly assumed that the authors were directly encoding the directional derivative as an infinite sum, which would offer limited improvements and result in a biased estimator—an approach already explored in prior literature. However, the authors clarified this point using the example of the $P_4$ estimator, demonstrating how *randomness in $n$* allows the sum to represent the directional derivative *in expectation*. This leads to an unbiased estimator for zeroth-order oracles using a finite number of function evaluations.
Given the involvement of random variables with complex dependencies, the authors appropriately employed more advanced analytical tools to rigorously justify their results. This careful treatment adds to the work’s credibility and technical depth. I now recognize this contribution as novel within the context of zeroth-order optimization. In particular, it opens the door to deriving high-probability bounds for zeroth-order estimators, which justifies my revised evaluation from a 4 (weak accept) to a 5 (accept).

Moreover, despite introducing randomness in $n$, the authors still provide performance guarantees that match those of existing zeroth-order methods. This highlights the practical viability of their approach, while also expanding the scope of zeroth-order optimization into the realm of unbiased stochastic optimization.

**Limitations:**

Yes. The authors have adequately addressed the limitations of this work.

**Quality:**

3

**Strengths And Weaknesses:**

**Strengths:**

1. The paper introduces a general framework for analyzing a class of zeroth-order estimators that estimate directional derivatives in an unbiased manner. This provides a unifying perspective on various estimation strategies.
2. The observation that an unbiased single-point estimator of the true gradient can have infinite variance—contrasting with the bounded variance of zeroth-order estimators for smoothed functions—is both surprising and insightful.

**Weaknesses:**

1. The contributions and practical implications of the work are not clearly articulated. The motivation for introducing this framework, as well as its advantages over existing methods, remains vague. Furthermore, the writing lacks a clear sense of direction regarding potential applications or future developments based on the proposed theory.
2. The proposed technique relies on the construction of a countably infinite set $p_n$ and corresponding smoothing parameters $\mu_n$ for $n = 1, \cdots, \infty$. It is not immediately clear how this formulation translates into a practical implementation. Further clarification on how this can be approximated or truncated in practice would be valuable. If truncation is necessary, then using standard smoothed gradient estimators—which are better understood and widely adopted—might be more practical and theoretically grounded.
3. The experimental setup is limited in scope. To better demonstrate the practical utility of the proposed method, it would be beneficial to include experiments on larger-scale models or more diverse benchmark tasks that highlight tangible improvements over existing approaches.

---

> ### Author Rebuttal · Authors · 2025-07-27
>
> We appreciate the reviewer’s constructive feedback. Below, we provide our point-by-point responses to each of questions raised.
>
> 1. (**how this observation contributes either theoretically or practically?  any concrete advantages?**) The result that unbiased estimators can attain the same level of variance in the smooth setting as standard single- or two-point estimators is certainly interesting. However, it remains unclear how this observation contributes either theoretically or practically. Specifically, does exact unbiasedness offer any concrete advantages over the existing analyses of zeroth-order optimization that rely on the gradient of a smoothed function?
>
>     **Response:**   We thank the reviewer for raising this important question. We highlight two key advantages and one potential advantage of using an unbiased estimator:
>
>     (i) **Practical benefit:** *First*, using the unbiased gradient estimator removes the need to balance bias and variance as a coupled trade-off. This simplifies the hyperparameter tuning process, allowing us to focus solely on controlling variance.  As indicated by our Theorem 3.2, the variance upper bounds of the $P_3$ and $P_4$ estimators are parameter-agnostic; that is, they do not introduce any additional hyperparameters compared to classical biased estimators. *Second*, our empirical results indicate that the reduced biased could benefit the gradient estimation, which could result in better performance in training neural network or other applications of zeroth-order optimization.
>
>     (ii) **Theoretical benefit:** We especially highlight the theoretical benefit of considering the unbiased gradient estimator. The zero bias enables the direct application of standard optimization theory without needing to account for the complications introduced by estimator bias. While our current analysis focuses on vanilla SGD (as in Corollary 3.5), the same theoretical framework readily extends to a broad range of optimization methods or learning theory results, in which the theoretical analysis only relies on the unbiased estimation of the true gradient.
>
>     (iii) **Potential benefit:** The unbiased estimator opens a novel and underexplored direction in the design of zeroth-order gradient estimators. While existing (typically biased) estimators have been extensively studied and refined, the space of unbiased estimators remains largely under-explored. We believe our work creates new opportunities for developing more effective and theoretically grounded zeroth-order estimators.
>
>     To illustrate the potential benefit (and the *'potential applications or future developments based on the proposed theory'* noted in the **Weakness 1**), we would like to clarify that in recent years, most zeroth-order gradient estimators have been constructed using the random smoothing technique. This approach approximates the objective function by a smoothed version of the form $\mathbb{E}_{v \sim V}[f(x + \mu v)]$. In contrast, our work is based on a fundamentally different principle: we express the gradient $\nabla f(x)$ as the expectation of a random variable that depends solely on function evaluations. Here we list two **potential future developments** based on this new expectation representation perspective (developed in our Section 2.1):
>     * > (Designing a new gradient estimator using frequency information): If we replace the random variable solely depending on function evaluations with the random variable depending on frequency of gradient (i.e. use Fourier series to replace the Taylor series), we can access the gradient using the information from the frequency domain. Our work can motivate a branch of further developments in using other side signals to recover the gradient.
>     * > (Reducing the memory usage of Adam-like optimizer): In Adam, the first- or second- order momentum is represented as a (truncated) series. If we replace it with the expectation representation (described in our Section 2.1), we can recover the whole trajectory using a single item of this series, avoding to save the momentum information on GPU.
>
>     We will include these future directions in our discussion and hope it can resolve your concern described in *Weakness 1*.
>
> 2. (**how selecting a countably infinite set of hyperparameters?**) The construction of these estimators involves selecting a countably infinite set of distributions and corresponding smoothing parameters . It is unclear how this construction guarantees exact unbiasedness in practice. Could this be interpreted instead as an approximation or a limiting form of unbiasedness? If so, why would this approach be preferable to, or more suitable than, using standard smoothed gradient estimators in algorithmic applications?
>
>     **Response:**   We appreciate the reviewer's insightful comment. These two parameters  $\\\{p_n\\\}$  and  $\\\{ \mu_n\\\}$ are indeed the main obstacle of implementing the unbiased zeroth-order gradient estimator, and our work exactly contributes to solve this problem.  These two sequences form an infinite-dimensional hyper-parameter space; by intuition, it must be hard to tune these parameters. However, we make a key observation derived in Theorem 3.2: if we require them to satisfy a simple algebraic relation; e.g. $p_n = \frac{\mu_n - \mu_{n+1}}{\mu}$ (for the $P_3$- and $P_4$-estimator), then it is guaranteed to achieve the optimal complexity. Moreover, the condition presented in our Theorem 3.2 is the necessary and sufficient condition. That is, if the presented relation is not satisfied, the variance must be sub-optimal (even though they may still achieve the optimal complexity in the asymptotic sense).  As the result, introducing these two parameters do not add additional difficulty to tune the hyper-parameter.
>
>     We also have included two examples (Example 3.3 and Example 3.4) to illustrate how to obtain $\\\{p_n\\\}$  and  $\\\{ \mu_n\\\}$  from this relation. To better illustrate its practical implementation, we further explain it here: Typically, we can select an existing distribution over $\mathbb{N}$ (e.g. the Geometric distribution or the Zipf's distribution) that satisfies (1) we can directly sample from using `numpy`, `scipy`, or `pytorch`, and (2) we can access the density function $p_n$ through  these packages. Then the estimation procedure is decribed as follows:
>     1. We sample $i \sim  \\\{p_n\\\}$ (the distribution we just selected) and calculate $\mu_i$ using our algebraic relation derived from our Theorem 3.2.
>     2. We plugging it back to our gradient estimator formula.
>
>     Then we directly use this estimator as the true gradient. The truncation occurs only if the numerical stability is absent by a specific $\mu_i$ (i.e. $\mu_i$ is too small). This could also happen for a classical zeroth-order estimation; though the classical method uses a constant $\mu$, at some point $x$ or for some data $\xi$, this specific $\mu$ could be too small. Therefore, the practical implementation doesn't present significant difference between the unbiased estimator and other gradient estimators. We also hope this clarification could resolve your concern presented in **Weakness 2**.
>
> We thank the reviewer again for these insightful comments. In summary, the goal of our work is to present a deeper understanding of the unbiased zeroth-order gradient estimator, which is a topic that has been rarely explored. From our perspective, we have successfully achieved our goal: (1) we identify the conditions under which constructing an unbiased estimator is possible; (2) we derive the optimal structure that achieves the best possible complexity; and (3) we determine the minimal number of function evaluations required to obtain an unbiased gradient estimator, improving the best-known result from 4 evaluations to 1. We sincerely hope the reviewer will take these theoretical insights into account when assessing our submission.

---

> > ### Comment · Reviewer_Yvd5 · 2025-08-02
> >
> > I would like to thank the authors for addressing my comments. I think that the significance of the work is much more clear to me now and with the response and discussions from the above reviews, I am able to get a rough idea of the implementational aspect of the work. I wanted to clarify one aspect of the theory (which I think is already mentioned in this work), so consider a $P(n, v)$ estimator, then it simply has dependance on $\mu_n$, $\mu_{n+1}$, and $\mu_1$ values only in terms of step-sizes, so if we are simply to use the distribution of these random variables as proposed then would the estimator be biased or unbiased for any $n$?

---

> > > ### Author Response · Authors · 2025-08-03
> > >
> > > We appreciate the reviewer's follow-up question. Before we present the answer, we hope clarify two potential approaches of employing the distribution of the random variable $n$ to make this question more clear:
> > >
> > > 1. **Formulation 1:** Fix the value of $n$ and use
> > > $$ \hat{\nabla} f(x) = P_4 (n,v) v =  \dfrac{ f(x+\mu_1 v) - f(x) }{ \mu_1 } v + \dfrac{1}{p_n}   \left[ \dfrac{ f(x+\mu\_{n+1} v) - f(x) }{\mu\_{n+1} } v - \dfrac{ f(x+\mu_n v) - f(x) }{ \mu_n } v \right]$$
> > > as the estimator. Here $p_n$ is selected as a known distribution e.g. the geometric distribution.
> > >
> > >     Then this estimator gives a **biased** estimator for arbitrary $n$. The bias comes from two sources: (i) the bias of the classical two-point estimator: $ \dfrac{ f(x+\mu_1 v) - f(x) }{ \mu_1 } v $; and (ii) the bias of the difference between two classical two-point estimators:  $ \dfrac{1}{p_n}   \left[ \dfrac{ f(x+\mu\_{n+1} v) - f(x) }{\mu\_{n+1} } v - \dfrac{ f(x+\mu_n v) - f(x) }{ \mu_n } v \right].$ As $n$ is fixed, we treat $p_n$ as a constant. If we apply the Taylor theorem to each $f(x+\mu v) $ in this estimator, then the bias would have the same level as the the classical two-point estimator $ \dfrac{ f(x+\mu_1 v) - f(x) }{ \mu_1 } v $ up to a few higher-order term relying on $\mu_1$.
> > >
> > > 2. **Formulation 2:** Use a full/truncated series as the estimator; that is,
> > > $$ \hat{\nabla} f(x) = \dfrac{ f(x+\mu_1 v) - f(x) }{ \mu_1 } v + \sum_n  \dfrac{1}{p_n}   \left[ \dfrac{ f(x+\mu\_{n+1} v) - f(x) }{\mu\_{n+1} } v - \dfrac{ f(x+\mu_n v) - f(x) }{ \mu_n } v \right].$$
> > >
> > >     Then this estimator gives a **unbiased** estimator (if we use a full series), or a **biased** estimator (if we simply omit everything after $N$-th item in this series).
> > >
> > >     For the truncated estimator, the bias would be dependent on the choice of $\\\{p_n\\\}$ and  $\\\{\mu_n\\\}$. If these hyper-parameters are poorly selected, then the bias could be possibly even worse than the standard estimator. From our perspective, the optimal selection to control this bias would be similar to the condition derived in our Theorem 3.2 but may have additional dependence on the truncation step $N$.
> > >
> > > ---
> > > We hope this addresses your concern regarding this setting. Please feel free to let us know if further clarification is needed.

---

> > > > ### Comment · Reviewer_Yvd5 · 2025-08-03
> > > >
> > > > Thank you for the prompt reply! This distinction helps clear some things up, but then my follow up question would be that how can one claim that we can get an unbiased estimator with just four function evaluations as mentioned in section 2.3? I have this confusion because it seems that if we use the full series, then we would be requiring an infinite number of function evaluations in this case, if I am not mistaken. Then, how does it answer your question in section 2.3 "What is the minimum number of function evaluations required to construct an unbiased gradient estimator?"?

---

> > > > > ### Author Response · Authors · 2025-08-03
> > > > >
> > > > > Thanks for the follow-up! We provide our further clarification below.
> > > > >
> > > > > ## **As the estimator is biased for each $n$, how does it become unbiased?**
> > > > >
> > > > > Let's start from the following estimator:
> > > > > $$  P_4(n,v)v = \dfrac{ f(x+\mu_1 v) - f(x) }{\mu_1 } v + \dfrac{1}{p_n}   \left[ \dfrac{ f(x+\mu\_{n+1} v) - f(x) }{\mu\_{n+1} } v - \dfrac{ f(x+\mu_n v) - f(x) }{ \mu_n } v \right]. $$
> > > > >
> > > > > We can make two observations here:
> > > > > 1. Observation 1: the value of $n$ is not fixed; instead, it is a random variable.
> > > > > 2. Observation 2: the bias in the difference term $ \dfrac{1}{p_n}   \left[ \dfrac{ f(x+\mu\_{n+1} v) - f(x) }{\mu\_{n+1} } v - \dfrac{ f(x+\mu_n v) - f(x) }{ \mu_n } v \right]$ could be either positive or negative.
> > > > >
> > > > > As the result, for some $n$, the difference term helps to reduce the bias from the first term; however, for some other $n$, the difference term will make the bias worse. Shortly speaking,
> > > > > > For some $n$, the estimator is better; for some $n$, the estimator is worse.
> > > > >
> > > > > But finally, we can find that, **in the in-expectation sense**, the bias is completely gone. To illustrate how the bias is gone, we make the following derivation:
> > > > > * Recall that $p_N := \mathbb{P}(n =N)$. We can calculate the expectation of $P_4(n,v)$ with respect to $n$: $\sum_N  \mathbb{P}(n =N) P_4(N,v). $ It is exactly
> > > > >
> > > > > $$ \dfrac{ f(x+\mu_1 v) - f(x) }{\mu_1 } + p_1 \times  \dfrac{1}{p_1}   \left[ \dfrac{ f(x+\mu\_{2} v) - f(x) }{\mu\_{2} } v - \dfrac{ f(x+\mu_1 v) - f(x) }{ \mu_1 } v \right] + p_2   \times  \dfrac{1}{p_2}   \left[ \dfrac{ f(x+\mu\_{3} v) - f(x) }{\mu\_{3} } v - \dfrac{ f(x+\mu_2 v) - f(x) }{ \mu_2 } v \right] + \dots$$
> > > > >
> > > > > * We calculate the first two terms. We find that $\dfrac{ f(x+\mu_1 v) - f(x) }{\mu_1 }$ is cancelled out:
> > > > >
> > > > > $$ \dfrac{ f(x+\mu_2 v) - f(x) }{\mu_2 }  + p_2   \times  \dfrac{1}{p_2}   \left[ \dfrac{ f(x+\mu\_{3} v) - f(x) }{\mu\_{3} } v - \dfrac{ f(x+\mu_2 v) - f(x) }{ \mu_2 } v \right]  + p_3   \times  \dfrac{1}{p_3}   \left[ \dfrac{ f(x+\mu\_{4} v) - f(x) }{\mu\_{4} } v - \dfrac{ f(x+\mu_3 v) - f(x) }{ \mu_3 } v \right] + \dots$$
> > > > >
> > > > > * Next, we find that $ \dfrac{ f(x+\mu_2 v) - f(x) }{\mu_2 }$ is cancelled out. We can repeat this calculation, and it results in a limit:
> > > > >
> > > > > $$ \lim_{n\to \infty}  \dfrac{ f(x+\mu_n v) - f(x) }{\mu_n }.$$
> > > > >
> > > > > This limit is exactly the directional derivative along the vector $v$; it is equal to $v^\top \nabla f(x)$. If the random vector is uniformly sampled from the sphere, then the estimator  $\lim_{n\to \infty}  \dfrac{ f(x+\mu_n v) - f(x) }{\mu_n } v = v v^\top \nabla f(x)$ is an unbiased estimator of the true gradient $\nabla f(x)$ (up to a scaling factor).
> > > > >
> > > > >
> > > > > ## **How does this answer "What is the minimum number of function evaluations required to construct an unbiased gradient estimator?"?**
> > > > >
> > > > > The above derivation is based on the four-point estimator $ P_4(n,v)$. Each time, we need to access four function values: $f(x), f(x+\mu_1 v), f(x+\mu_n v), f(x+\mu\_{n+1} v) $. Let us go back to this estimator:
> > > > >
> > > > > $$  P_4(n,v)v = \dfrac{ f(x+\mu_1 v) - f(x) }{\mu_1 } v + \dfrac{1}{p_n}   \left[ \dfrac{ f(x+\mu\_{n+1} v) - f(x) }{\mu\_{n+1} } v - \dfrac{ f(x+\mu_n v) - f(x) }{ \mu_n } v \right]. $$
> > > > >
> > > > > Before we take the expectation with respect to $n$, we can construct another estimator to estimate $P_4(n,v)$.  The new estimator should satisfy two property:
> > > > >
> > > > > * (i) It is equal to $P_4(n,v)$ (in the in-expecation sense), and
> > > > > * (ii) it only requires $1$ function evaluation.
> > > > >
> > > > > The construction could be given by a specific sampling procedure over the required function values $f(x), f(x+\mu_1 v), f(x+\mu_n v), f(x+\mu\_{n+1} v) $, which leads to a specific construction $P_1(n,v)$ given in Equation (9), Section 2. Here, we also note that this estimator indeed depends on one more random variable (instead of just $n$). In Equation (9), this random variable is denoted as $U_4 $ (called the selection random variable, as it selects which point to evaluate).
> > > > >
> > > > > To answer the original question  "What is the minimum number of function evaluations required to construct an unbiased gradient estimator?", as the estimator $P_1(n,v)$ only requires exactly $1$ function evaluation, we can conclude that this  minimum value is $1$.
> > > > >
> > > > >
> > > > > ---
> > > > >
> > > > > We sincerely appreciate the reviewer’s thoughtful engagement. Please let us know if any part of our response remains unclear or if there are specific points that would benefit from further clarification.

---

> ### Comment · Reviewer_Yvd5 · 2025-08-03
> **Raise score from 3 to 5**
>
> I would sincerely like to thank the authors for clearing out my concerns. This completely clears out all doubts on my end. I apologize for the mistake on my end. I did not realize that $n$ itself is a random variable, which in case would imply that $n$ has infinite support not an infinite number of parameters, as mistakenly stated in my earlier comment. I think that this work is novel and very interesting now and would like to raise my score from a 3 to a 5.

---

### Official Review · Reviewer_Ea9x · 2025-06-22

**Clarity:** 4
**Significance:** 2
**Originality:** 2
**Rating:** 4
**Confidence:** 3

**Summary:**

The central contribution of this paper is the introduction of a novel family of provably unbiased zeroth-order gradient estimators, which the authors term Pk-estimators (where k = 1, 2, 3, 4 denotes the number of function queries required). A detailed variance analysis reveals that while the P1 estimator may exhibit infinite variance, the P2, P3, and P4 estimators are shown to possess finite variance under certain conditions.

**Questions:**

Clarification is requested regarding the plots in Figure 2. The x-axis of the right panel is labeled "function calls," while the left panel shows the "number of updates." For the one-sided and two-sided two-point estimators, one would expect the total number of function calls to be twice the number of gradient updates. However, the plots for these estimators conclude at approximately 80,000 function calls respectively, rather than the expected 40,000 corresponding to 20,000 updates. Could you please explain this apparent discrepancy? If I am wrong, please correct me.

**Ethical Concerns:**

["NO or VERY MINOR ethics concerns only"]

**Final Justification:**

The author presents an unbiased estimate of the gradient of the original function, which is mathematically meaningful; however, its contribution to the machine learning community remains subject to debate. Moreover, the rebuttal addresses most of my concerns. Therefore, I have decided to raise my score.

**Quality:**

2

**Strengths And Weaknesses:**

- The paper's objective is clearly articulated, and the manuscript is well-written, making the core concepts and contributions accessible.

- The novelty of the proposed technique is a point of concern, as it appears to share similarities with methods presented in Chen (2023). Furthermore, a more robust justification for the pursuit of an unbiased estimator is needed. Given that the resulting optimization algorithm demonstrates a convergence rate comparable to methods using biased estimators, what is the fundamental advantage of this approach? Why should one prefer this unbiased estimator over a simpler, biased alternative?

---

> ### Author Rebuttal · Authors · 2025-07-27
>
> We deeply appreciate the reviewer’s constructive feedback. Below, we provide our point-by-point responses to each of the concerns raised.
>
> ## Responses to weakness
>
> - (**Similarity to Chen (2020)**) The novelty of the proposed technique is a point of concern, as it appears to share similarities with methods presented in Chen (2023).
>
>     **Response:** The telescoping structure is the same as the one used in Chen (2020); however, our contribution goes much beyond this telescoping structure and we provide a deeper understanding to the unbiased zeroth-order gradient estimator:
>
>     1.  **Identify when we can use unbiased gradient estimator:** We identify the condition under which the telescoping structure admits a valid expectation representation (Proposition 2.1). This condition is critical for constructing unbiased estimators, but has never been established in Chen (2020) or other prior work.
>
>     2. **More general unbiased gradient estimator \& Reduce the number of function evaluations from 4 to 1:** The estimator in Chen (2020) is a special case of our $P_4$-estimator. Our framework extends beyond this, answering a fundamental theoretical question: *What is the minimal number of function evaluations needed to construct an unbiased gradient estimator?* We improve the known answer from 4 give by Chen (2020) to 1. Although our one-point estimator comes with a negative result (this is also a new result), we believe it remains of independent theoretical interest.
>
>     3. **Identify the necessary and sufficient condition of achieving the optimal variance:** Moreover,  one of our key focuses is identifying optimal parameter sequences $\\\{\mu_n\\}$ and $\\{p_n\\\}$ (Theorem 3.2), rather than proposing a specific estimator. This result indicates that we cannot simply take an arbitrary sequence as being done in Chen (2020), instead, we need to require them to follow a specific algebraic structure.
>
>     These substantial contributions are beyond the shared telescoping structure between Chen (2020) and our work, and we sincerely hope the reviewer will take these theoretical insights into account when assessing our submission.
>
>
>
> - (**Advantage compared to biased estimator**) Furthermore, a more robust justification for the pursuit of an unbiased estimator is needed. Given that the resulting optimization algorithm demonstrates a convergence rate comparable to methods using biased estimators, what is the fundamental advantage of this approach? Why should one prefer this unbiased estimator over a simpler, biased alternative?
>
>     **Response:** We thank the reviewer for raising this important question. We highlight two key advantages and one potential advantage of using an unbiased estimator:
>
>     (i) **Practical benefit:** *First*, using the unbiased gradient estimator removes the need to balance bias and variance as a coupled trade-off. This simplifies the hyperparameter tuning process, allowing us to focus solely on controlling variance.  As indicated by our Theorem 3.2, the variance upper bounds of the $P_3$ and $P_4$ estimators are parameter-agnostic; that is, they do not introduce any additional hyperparameters compared to classical biased estimators. *Second*, our empirical results indicate that the reduced biased could benefit the gradient estimation, which could result in better performance in training neural network or other applications of zeroth-order optimization.
>
>     (ii) **Theoretical benefit:** The zero bias enables the direct application of standard optimization theory without needing to account for the complications introduced by estimator bias. While our current analysis focuses on vanilla SGD (as in Corollary 3.5), the same theoretical framework readily extends to a broad range of optimization methods or learning theory results, in which the theoretical analysis only relies on the unbiased estimation of the true gradient.
>
>     (iii) **Potential benefit:** The unbiased estimator opens a novel and underexplored direction in the design of zeroth-order gradient estimators. While existing (typically biased) estimators have been extensively studied and refined, the space of unbiased estimators remains largely under-explored. We believe our work creates new opportunities for developing more effective and theoretically grounded zeroth-order estimators.
>
> ## Responses to questions
>
> - (**Figure 2 discrepancy: function calls vs. updates seem mismatched**) Clarification is requested regarding the plots in Figure 2. The x-axis of the right panel is labeled "function calls," while the left panel shows the "number of updates." For the one-sided and two-sided two-point estimators, one would expect the total number of function calls to be twice the number of gradient updates. However, the plots for these estimators conclude at approximately 80,000 function calls respectively, rather than the expected 40,000 corresponding to 20,000 updates. Could you please explain this apparent discrepancy? If I am wrong, please correct me.
>
>     **Response:** We appreciate the reviewer's careful reading. We would like to clarify that the x-axis is correctly labeled here, as we are using the mini-batch two-point gradient estimator to align the number of function evaluations to $4$; that is, we increase the batch size to ensure that each gradient update takes the same number of function evaluations as the $P_4$-estimator.  More explicitly, the exact number is described as the following:
>     - For the two-side two-point estimator, we evaluate four points: $f(x+\mu v_1)$, $f(x-\mu v_1)$. $f(x+\mu v_2)$, $f(x-\mu v_2$). Then we estimate the gradient using $\frac{1}{2}[\frac{f(x+\mu v_1) - f(x-\mu v_1)}{2 \mu}v_1+ \frac{f(x+\mu v_2) - f(x-\mu v_2)}{2 \mu}v_2]$. It corresponds to the batch size $b=2$.
>     - For the one-side two-point estimator, we evaluate four points: $f(x)$, $f(x+\mu v_1)$, $f(x+\mu v_2)$, $f(x+\mu v_3)$ Then we estimate the gradient using $\frac{1}{3}[\frac{f(x+\mu v_1) - f(x)}{ \mu}v_1+ \frac{f(x+\mu v_2) - f(x)}{ \mu}v_2+ \frac{f(x+\mu v_3) - f(x)}{ \mu}v_3]$. It corresponds to the batch size $b=3$.
>
>     We will also update the figure to include the case of batch size $b=1$ in the revision for a direct comparison.

---

> > ### Comment · Reviewer_Ea9x · 2025-08-01
> > **Reply to rebuttals**
> >
> > I am curiously about the experiments result about $b=1$. Can you show one table here?

---

> > > ### Author Response · Authors · 2025-08-01
> > >
> > > Absolutely! Below we present the table-form trajectory of our language model optimization experiment, corresponding to Figure 2 in our submission. Specifically, we include two tables: *(i)* Training Loss vs. Number of Updates (left panel of Figure 2), and *(ii)* Training Loss vs. Number of Function Calls (right panel of Figure 2). The last four rows are directly taken from the results reported in our submission.  The methods marked with an asterisk (*) represent the classical two-point estimator
> > >
> > > * (One-Side Two-Point Estimator) $\hat{\nabla}f(x)=\frac{1}{b}\sum\_{i=1}^b \dfrac{f(x+\mu v\_i) - f(x)}{\mu} v\_i$
> > > * (Two-Side Two-Point Estimator) $\hat{\nabla}f(x)=\frac{1}{b}\sum\_{i=1}^b \dfrac{f(x+\mu v\_i) - f(x-\mu v\_i)}{2\mu} v\_i$
> > >
> > > with the batch size $b=1$.
> > >
> > >
> > >
> > > | Estimators\Number of Updates                 | 5000   | 10000  | 15000  | 20000  |
> > > | -------------------------------------------- | ------ | ------ | ------ | ------ |
> > > | First-Order Adam                             | 0.0500 | 0.0135 | 0.0060 | 0.0034 |
> > > | First-Order SGD                              | 0.4642 | 0.3453 | 0.2855 | 0.2475 |
> > > | **(*) One-Side Two-Point Estimator ($b=1$)** | 0.5258 | 0.4614 | 0.3825 | 0.3843 |
> > > | **(*) Two-Side Two-Point Estimator ($b=1$)** | 0.5208 | 0.4521 | 0.4068 | 0.3720 |
> > > | One-Side Two-Point Estimator ($b=3$)         | 0.5202 | 0.4375 | 0.3623 | 0.3498 |
> > > | Two-Side Two-Point Estimator ($b=2$)         | 0.5200 | 0.4489 | 0.3739 | 0.3165 |
> > > | Geometric $P_3$-Estimator                    | 0.4517 | 0.3752 | 0.3438 | 0.2895 |
> > > | Geometric $P_4$-Estimator                    | 0.3873 | 0.3154 | 0.2898 | 0.2657 |
> > >
> > >
> > >
> > > | Estimators\Number of Function Calls          | 20000  | 40000  | 60000  | 80000  |
> > > | -------------------------------------------- | ------ | ------ | ------ | ------ |
> > > | **(*) One-Side Two-Point Estimator ($b=1$)** | 0.4614 | 0.3843 | -      | -      |
> > > | **(*) Two-Side Two-Point Estimator ($b=1$)** | 0.4521 | 0.3720 | -      | -      |
> > > | One-Side Two-Point Estimator ($b=3$)         | 0.5202 | 0.4375 | 0.3623 | 0.3498 |
> > > | Two-Side Two-Point Estimator ($b=2$)         | 0.5200 | 0.4489 | 0.3739 | 0.3165 |
> > > | Geometric $P_3$-Estimator                    | 0.4408 | 0.3874 | 0.2895 | -      |
> > > | Geometric $P_4$-Estimator                    | 0.3873 | 0.3154 | 0.2898 | 0.2657 |
> > >
> > >
> > >
> > > ### Discussion on the results
> > >
> > > The result is as expected. Choosing larger batch sizes gives more accurate gradient estimates, leading to lower training loss when measured by the number of updates. However, we also observe that when comparing methods under a fixed number of function calls, selecting the batch size as  $b=1$  also presents its own advantage. As the result, choosing the batch size in zeroth-order optimization is non-trivial and it requires to balance the variance of gradient estimation against the per-step cost.
> > >
> > > ---
> > >
> > > We especially thank the reviewer for drawing our attention to this important yet previously overlooked issue. We will update the figure and add discussions in the revision.

---

> > > > ### Comment · Reviewer_Ea9x · 2025-08-02
> > > > **Reply to rebuttals**
> > > >
> > > > The rebuttal addresses most of my concerns. Therefore, I have decided to raise my score.

---

### Official Review · Reviewer_nG3r · 2025-06-26

**Clarity:** 4
**Significance:** 3
**Originality:** 3
**Rating:** 5
**Confidence:** 4

**Summary:**

The paper proposes new  unbiased estimates of the directional derivative, based on a telescopic series representation of the directional derivative. The paper is very well written, clear, and the proofs are easy to follow. My only real concern is how this work fits within Neurips. I understand there is still plenty of legacy code (often written in Fortran) for simulations that requires zero order training. But here the connection to Neurips + ML is through the proposed application of model fine-tuning, and to save of memory when using autodiff. But here I have a question/issue. See [Application to LLM fine-tuning] below. Based on the response of the authors on this question, and the other reviewers, I am open to reconsidering my score.

**Questions:**

1. The reference lower bound in [Duchi et al. 2015] is for zero-order optimization, and not first-order. The fact that you have an unbiased estimator, means you can leverage the proof structure of SGD, such as the one in [Khaled and Richtarik]. Because of this, your statements in the main text and abstract such as "... prove that SGD using the proposed estimators achieves optimal complexity for smooth non-convex objectives" can be misleading. A reader could interpret that you have matched the optimal complexity of "SGD for smooth non-convex objectives" which is


$$\frac{ \sqrt{2} L (f(x^0) - \inf f) + (\inf f - \mathbb{E}[\inf f\_{\xi}])}{\sqrt{T}} $$

see Theorem 5.12 in
Garrigos et. al, Handbook of Convergence Theorems for (Stochastic) Gradient Methods, arXiv, 2024

I think you should re-word this contribution, to make it clear you are talking about the optimal zero-order complexity.

2. How do your methods compare to other zero order methods used for fine-tuning, such as Mezo in [Malladi et al 2023]? Or is Mezo equivalent here to what you call Two-side point estimator? On this note, I see a small mistake in your Two-side point estimator in Table 1. You are off by a factor of 2, since it should be
$$ \frac{f(x+\mu v_i)- f(x+\mu v_i)}{2\mu} v_i$$
This same mistake is also repeated on page 14 in three locations.
A factor of two would make no difference if you tuned the learning rate. But since you have a fixed $\eta = 10^{-4}$, this factor could make a difference.



[Application to LLM fine-tuning]. To justify this application, you have cited some prior work, and stated "storing the full computational graph ... can be prohibitively large for modern language models". I checked your references, and only one [Malladi et al 2023] has been peer reviewed. There, their claims is the using zero-order uses a fraction of the memory of backpropagtion. This I agree with, since backpropagation requires storing all weights, and adjoints of your computational graph. But here is where I see an issue. These zero order methods should instead be compared to the cost/memory of computing the directional derivative $\nabla_v f(x)$ when using forward propagation, which can be done with a very small memory overhead. To justify the case of using zero-order for fine-tuning, you would need to compare to using forward prop for computing $\nabla_v f(x)$, and compare both the time, memory usage, and the resulting performance. Otherwise I don't think it is a compelling argument to apply these methods to fine-tuning.

### Forward-Mode AD: Flow and Memory Usage

In forward-mode automatic differentiation, to compute the directional derivative $\nabla f(x)^T d $, we propagate both the primal values $ x_\ell $ and tangent vectors $\dot{x}_\ell $ through each layer of the function composition $f = f_L \circ \cdots \circ f_1 $:

$x_{\ell} = f_{\ell}(x_{\ell-1}),$

$\dot{x}\_\ell = J\_\ell(x\_{\ell-1}) \dot{x}\_{\ell-1}$

where $J\_{\ell}$ is the Jacobian of the $\ell$th layer.
**Memory usage** is minimal: only the current pair $(x_{\ell-1}, \dot{x}_{\ell-1})$ must be stored to compute the next. Both values and tangents can be discarded once the next layer is evaluated, making forward-mode memory-efficient for directional derivatives.


**Minor points**

1. Line 625, if $a_n = \mu_{n+1}-\mu_n$ then $\sum_{n=1}^{\infty} a_n = -\mu_1$ instead of $\mu_1.$ The proof here is nonetheless correct

**Ethical Concerns:**

["NO or VERY MINOR ethics concerns only"]

**Final Justification:**

The authors engaged with my main issue, which was including forward mode differentiation as a baseline. They found limitations when using forward mode (I.e flash attention), were transparent on the potential benefits of using forward mode differentiation and promised to include forward mode in their background material. I was very pleased with open and scientific attitude of the authors, and also as mentioned in my initial review, see many merits in the paper. So I’m increasing my score to an accept.

**Limitations:**

No explicit mention of limitations.

**Paper Formatting Concerns:**

No clear issues.

**Quality:**

4

**Strengths And Weaknesses:**

The paper is very clear, the quality of the writing, and proofs is high. The unbiased zero order estimators are mostly new, excluding a 2020 paper by Chen.

---

> ### Author Rebuttal · Authors · 2025-07-27
>
> We deeply appreciate the reviewer’s careful reading and constructive feedback. We have fixed the minor issues and typos raised by the reviewer. Below, we provide our point-by-point responses to each of the questions raised.
>
> 1. **Misleading statement: “optimal complexity for smooth non‑convex objectives” could be read as first‑order optimality.**
>
>     **Response:** We appreciate the reviewer's suggestion. We agree that our sentence here is indeed misleading. We will revise it by correcting the misuse of citing  [Duchi et al. 2015] from the first-order to the zeroth-order optimization lower bound and re-wording the statement of applying [Khaled and Richtarik]. The revised paragraph is now given as the follow:
>
>     > (Revised paragraph) This complexity has matched the lower bound of solving a smooth non-convex optimization problem using **zeroth-order** gradient-based method [Duchi et al., 2015] and cannot be further improved without adding additional assumptions. **Though we directly apply the result from Khaled and Richtárik [2022] (which is applicable for all unbiased estimators), the zeroth-order estimation can result in an additional dependence on the dimension $d$; this dependence has been reflected in our upper bound.**
>
> 2. **(1) Comparison to other zero-order methods used for fine-tuning, such as Mezo in [Malladi et al 2023].**
>
>     **(2) Off a factor $2$ in the two-point estimator.**
>
>     **Response:** We appreciate the reviewer's careful reading and insightful comments.
>
>     - **Regarding the MeZO algorithm**, MeZO is not a new gradient estimator; instead, it is a memory-efficient implementation of the zeroth-order SGD to make existing gradient estimator (e.g. the two-side two-point estimator) a feasible approach to fine-tune LLMs. Before MeZO, people may enumerate each layer and assign the estimated gradient to `param.grad`. After the assignment is done, people take `optimizer.step()` to update the parameter exactly as the first-order method does. The MeZO paper has a smart observation: it is not necessary to save the estimated gradient to `param.grad`; instead, it suffices to save the random seed and the function value difference $[f(x+\mu v)-f(x-\mu v)]/(2\mu)$; when we need to recover the gradient (for the gradient update of each layer), we just need to use the random seed to re-generate the random vector $v$. This approach will significantly reduce the memory cost of the zeroth-order optimization approach.
>
>         As the result, our two-side two-point estimator is the same as the one used in the MeZO paper (with using the zeroth-order batch size $b=2$; we also note that in the original MeZO paper, $b$ is set to be $1$).
>
>     - **Regarding the omitted factor $2$**, we appreciate the reviewer's careful reading; it is fortunate that it is just a typo, and in our experimental implementation (located in the class given below), we have correctly included this factor.
>
>         - The synthetic experiment: `synthetic-experiments/exp_dim_comparison.ipynb`: the `CentralizedUniformEstimator` class.
>         - The LLM training experiment: `llm-fine-tuning/zo-bench/trainer.py`: the `zo_step` method in the `OurTrainer` class.
>
>         We will fix this typo by adding the factor $2$ in the table and on page 14.
>
> 3. **Forward‑mode AD may achieve directional derivatives with low memory and better performance.**
>
>     **Response:** We appreciate the reviewer's thoughtful input. We enjoy reading this comment and learn a lot from it. We agree with the reviewer that there could be other memory-efficient approach (e.g. Forward-mode AD) that could be better than the existing zeroth-order optimization approach. We would be glad to add a discussion of this topic in the background.
>
>     As the goal of our paper is to present a deeper understanding of the unbiased zeroth-order gradient estimator, it is indeed out of the scoop of our paper to include experiments for comparing the performance of Forward-mode AD with the zeroth-order optimization approach on LLM fine-tuning applications. Here we would like to recap our theoretical contributions: (1) we identify the conditions under which constructing an unbiased estimator is possible; (2) we derive the optimal structure that achieves the best possible complexity; and (3) we determine the minimal number of function evaluations required to obtain an unbiased gradient estimator, improving the best-known result from $4$ evaluations [Chen2020] to $1$. We sincerely hope the reviewer will take these theoretical insights into account when assessing our submission.

---

> > ### Comment · Reviewer_nG3r · 2025-08-01
> > **Thank you, still concerned with point 3**
> >
> > Thank you for your response regarding point 1 and 2. My main question, which is, why are you using zero order over forward AD does still stand. To be clear, I do appreciate your theoretical contributions and the points you raised. Your paper would be a strong contribution in the right venue, such as an optimization journal focused on zero order optimization, or areas where simulation based inference is important, where only function evaluation are available.
> >
> >  But in fine tuning a deep learning model, one can use forward AD to compute exactly the directional derivative, with almost no additional memory footprint. So why use a zero order approach? Because of this, I just don't see why this should be published at Neurips. I will continue to follow responses here, and any resulting discussions.

---

> > > ### Author Response · Authors · 2025-08-01
> > >
> > > Thank you for the continued engagement and immediate follow-up. We sincerely appreciate your recognition of our theoretical contributions and are especially grateful for the opportunity to further discuss the Forward AD approach.
> > >
> > > ##  **Our understanding on the forward AD approach**
> > >
> > > To ensure we are on the same page, we would like to briefly summarize our understanding of the Forward AD approach. If there are any inaccuracies in our interpretation, we welcome your corrections.
> > >
> > > The forward gradient provides the **exact** directional derivative (with exactly zero bias), while the zeroth-order approach offers only an **approximation** of the derivative gradient. More explicitly:
> > > * The forward AD approach calculates the directional derivative along the direction $v$, denoted by $v^\top \nabla f(x)$; and then it uses $vv^\top  \nabla f(x)$ as an estimation of the true gradient $\nabla f(x)$.
> > > * The zeroth-order optimization (including both our approach and the classical two-point estimation) is using an estimator of the directional derivative to replace $v^\top  \nabla f(x)$.
> > >
> > > As a result, the zeroth-order approximation inherently introduces additional variance (even if it can be unbiased). This makes the Forward AD method theoretically better in terms of estimator quality.
> > >
> > > ## **When is the zeroth-order approach preferable?**
> > >
> > > Here we focus on scenarios where Forward AD is accessible (e.g., fine-tuning LLMs), as zeroth-order optimization is clearly advantageous in black-box settings where the forward gradient is not available.
> > >
> > > *  **Implementation Difficulty:** The practical implementation of Forward AD heavily relies on the availability of JVP (a.k.a. the Jacobian-Vector Product). A naive implementation will not reduce the memory usage and potentially increase the computation cost. In some modern machine learning frameworks (for example, the FlashAttention), the JVP is not supported. As commented by the member of the open-sourced project `flash-attention` in the issue "JVP support for FlashAttention #1672":
> > >     > "JVP would be nontrival to implement".
> > >
> > >     Thus, in contexts where implementing JVP is complex or infeasible, the zeroth-order approach offers a more accessible alternative.
> > >
> > > * **Memory usage:** We fully agree with the reviewer’s observation that Forward AD can be memory-efficient when implemented properly. Specifically, if intermediate pairs are discarded immediately after use, the peak memory is at most $\max_\ell (|x_\ell |+  |a_\ell|)$, where $|x_\ell|$ is the size of $\ell$-th layer's parameter and $|a_\ell|$ is the size of the JVP vector.  However, the zeroth-order approach can further reduce the memory usage to $\max_\ell |x_\ell |$, as the zeroth-order approach doesn't require the JVP. As the result, the Forward AD approach still presents a higher memory usage.
> > >
> > >     Therefore, for the edge device or other extreme cases where the memory cost is sensitive, we may prefer the zeroth-order approach.
> > >
> > > ---
> > > In summary, we have identified two scenarios where the zeroth-order method is preferable. We hope this clarifies that the Forward AD and zeroth-order approaches are not mutually exclusive, but **complementary**,  even in settings where directional gradients are available.

---

> > > > ### Author Response · Authors · 2025-08-07
> > > >
> > > > Dear Reviewer nG3r,
> > > >
> > > > Thank you for taking the time to thoughtfully evaluate our paper. As the author-reviewer discussion period is ending soon and we have not yet heard back from you since our last reply, we wanted to kindly follow up to see if our rebuttal has addressed the concerns you raised in your review.
> > > >
> > > > We would be more than happy to continue the discussion if you have any further questions or comments.
> > > >
> > > > Sincerely,
> > > > The Authors

---

> > > > > ### Comment · Reviewer_nG3r · 2025-08-08
> > > > > **Point taken, reconsidering**
> > > > >
> > > > > Dear authors,
> > > > >
> > > > > I’m sorry about the delay in replying, but I do appreciate your latest response, and confirm that jvp is not implemented for flashattention. So at the very least, it is not a given that jvp is always implemented.
> > > > >
> > > > > I have also reached out to one of the authors of the mezo paper. Though I don’t see this in their paper, they report having done experiments benchmarking against forward model diff, and state that noisy zero order estimators in fact resulted in a better fine tuned model, as compared to exact forward mode diff.
> > > > >
> > > > > I’m seriously considering increasing my score, but would really like to see  a proper discussion around this point, and even if possible an experiment. If what this author of mezo says is true, having a clear experiment to confirm it would have real value, and opens up interesting questions.

---

> > > > > > ### Author Response · Authors · 2025-08-08
> > > > > >
> > > > > > Thank you for the follow-up and for sharing this valuable information!
> > > > > >
> > > > > > We agree that developing a theoretical explanation for why the noisy zeroth-order approach can sometimes outperform the exact forward gradient method would be an interesting and worthwhile direction for future research.
> > > > > >
> > > > > > In the meantime, we refer to the empirical results reported by [Zhang et al., 2024], which present a comparison between Forward AD and the Zeroth-Order approach:
> > > > > >
> > > > > > * [Zhang et al., 2024] Zhang, Yihua, et al. "Revisiting Zeroth-Order Optimization for Memory-Efficient LLM Fine-Tuning: A Benchmark." International Conference on Machine Learning. PMLR, 2024.
> > > > > >
> > > > > > **Performance Comparison (Sometimes Better, Sometimes Worse):**
> > > > > >
> > > > > > In the above paper, the performance comparison is shown in  *Table 2* and *Table 3*, where the Forward AD approach is referred to as Forward-Grad. We present their Table 3 result below:
> > > > > >
> > > > > > |            | OPT-13B | LLaMA2-7B | Vicuna-7B | Mistral-7B |
> > > > > > |------------|---------|-----------|-----------|------------|
> > > > > > | **COPA**   |         |           |           |            |
> > > > > > | Forward-Grad | **89**      | 82        | **84**        | 88         |
> > > > > > | ZO-SGD       | 87      | **86**        | 83        | **90**         |
> > > > > > | **WinoGrande** |       |           |           |            |
> > > > > > | Forward-Grad | **62.9**    | 64.3      | 65.6      | **70.1**       |
> > > > > > | ZO-SGD       | 62.6    | 64.3      | 65.6      | 68.7       |
> > > > > >
> > > > > > Contrary to the claim made by the authors of the MeZO paper, the results appear mixed: in some cases, the Forward AD method performs better, while in some other cases, the zeroth-order approach performs better.  Based on these observations, it may be difficult to design a definitive experiment to confirm the claim, as it does not appear to hold universally.
> > > > > >
> > > > > > ---
> > > > > > We thank the reviewer again for the constructive discussion and valuable feedback, and we hope that the above empirical comparison from the existing literature helps address the reviewer’s concerns.

---

### Official Review · Reviewer_5co9 · 2025-06-30

**Clarity:** 3
**Significance:** 3
**Originality:** 3
**Rating:** 5
**Confidence:** 4

**Summary:**

Authors propose new way to build unbiased estimate of gradient via gradient-free setup. The idea is interesting and according to authors claim propose for free a way to eliminate bias (by conserving the variance estimate).

**Questions:**

Why do authors use such a finite difference? It seems that symmetric one will be better
Shamir O. An optimal algorithm for bandit and zero-order convex optimization with two-point feedback //Journal of Machine Learning Research. – 2017. – V. 18. – №. 52. – P. 1-11.

**Ethical Concerns:**

["NO or VERY MINOR ethics concerns only"]

**Final Justification:**

I consider the paper will be useful for gradient-free community, though in initial variant in some places the paper was not accurate in interpretation of details. Now authors fix this drawbacks that is why my score is 5.

**Limitations:**

Main limitation-- if $f$ is available with noise like in Gasnikov A. et al. The power of first-order smooth optimization for black-box non-smooth problems //International Conference on Machine Learning. – PMLR, 2022. – V. 7241-7265. In this case the construction seems to be quite sensitive and may work worse than classical one approach.

**Paper Formatting Concerns:**

No problems

**Quality:**

3

**Strengths And Weaknesses:**

So the paper is well written and propose a useful way to reduce bias. But I'm not sure that this way leads to the same variance at the end as it will be for biased case, see e.g. Tsybakov A. et al. http://www.jmlr.org/papers/volume25/23-0733/23-0733.pdf or Gasnikov A. V. et al. Stochastic online optimization. Single-point and multi-point non-linear multi-armed bandits. Convex and strongly-convex case // Automation and remote control. – 2017. – V. 78. – P. 224-234. Roughly speaking, for me it seems that $d^3\mu^2$ terms could be improved as $d^2\mu^2$. May be it's worth to include more accurate discussion about this.

---

> ### Author Rebuttal · Authors · 2025-07-27
>
> We deeply appreciate the reviewer’s positive evaluation and constructive feedback. Below, we provide our point-by-point responses to each of the concerns raised.
>
> ## Response to weakness
>
> - (**Include more accurate discussions on the variance term $d^3 \mu^2$ vs. $d^2 \mu^2$**) So the paper is well written and propose a useful way to reduce bias. But I'm not sure that this way leads to the same variance at the end as it will be for biased case, see e.g.
>
>     - Akhavan, Arya, et al. "Gradient-free optimization of highly smooth functions: improved analysis and a new algorithm." *Journal of Machine Learning Research* 25.370 (2024): 1-50.
>     - Gasnikov, Alexander V., et al. "Stochastic online optimization. Single-point and multi-point non-linear multi-armed bandits. Convex and strongly-convex case." *Automation and remote control* 78.2 (2017): 224-234.
>
>     Roughly speaking, for me it seems that $d^3 \mu^2$ terms could be improved as $d^2 \mu^2$. May be it's worth to include more accurate discussion about this.
>
>     (**Function value noise**) If is available with noise like in:
>
>     - Gasnikov, Alexander, et al. "The power of first-order smooth optimization for black-box non-smooth problems." *International Conference on Machine Learning*. PMLR, 2022.
>
>     In this case the construction seems to be quite sensitive and may work worse than classical one approach.
>
> - **Response:** We deeply appreciate the reviewer's insightful comment; it indeed identifies an important setup of zeroth-order optimization which we lack sufficient discussion. When considering the scenario where the function evaluation is exact (considered in our setting), the variance error term contains $d^3 \mu^2$ is worse than $d^2 \mu^2$ but it could be resolved by setting a sufficiently small $\mu$. So, it still leads to the optimal complexity. However, in the noisy oracle setting (as studied in above references), this term will have negative impact on final result. These works spend additional effort to delivery a tighter characterization on the upper bound. To better clarify the difference between these settings, we will add the following discussion right after our current Comparison with Existing Literature.
>
>     > (Comparison with the Noisy Oracle Setup)  In our work, we consider the exact function evaluation setting with noiseless values. In this case, our variance scales as $d^3 \mu^2$, which is worse than the $d^2 \mu^2$ of some specific biased estimators. However, this can be mitigated by choosing a small enough $\mu$, so the overall sample complexity remains optimal. However, in the noisy function evaluation setting, where each function evaluation may return a noisy value, a smaller $\mu$ amplifies the noise, leading to degraded performance. Several recent works have provided more refined analysis under noisy setups with improved variance behavior. Notably, Akhavan et al. (2024) demonstrated that for highly smooth functions, the $\ell_1$-randomization can reduce the variance scaling to $d^2 \mu^2$ with achieving the improved performance for highly smooth objective functions, which extends the existing $\ell_1$-randomization proposed by Akhavan et al. (2022). Earlier foundational work by Gasnikov et al. (2017) analyzed the variance behavior in single-point and multi-point bandit feedback settings, and more recent developments further explore the impact of first-order smoothness in noisy black-box optimization [Gasnikov et al., 2022]. Notably, all of these results achieve the optimal complexity derived by Duchi et al., 2015.
>
> ## Response to questions
>
> - Why do authors use such a finite difference? It seems that symmetric one will be better according to [Shamir2017].
>
>     **Response:**  We appreciate the reviewer’s insightful comment. We agree that the symmetric one (also called the two-side two-point estimator) is commonly considered better than the naive one-side two-point estimator. However, there are two reasons for still considering the one-side estimator:
>     (1) On the theoretical side, they still achieve the same sample complexity for smooth objective functions. This result is implied by the lower bound results in [Duchi2015].
>     (2) For the practical consideration, while the symmetric estimator uses $f(x+\mu v)$ and $f(x-\mu v)$ to compute the gradient, it commonly requires evaluating $f(x)$ for logging purposes. This operation will introduce an additional function evaluation.
>
>     Moreover,  our choice is motivated by a deeper theoretical interest in a relatively underexplored class of estimators characterized by structured hyperparameters (i.e. $\{\mu_n\}$ and $\{p_n\}$). These hyper-parameters do not appear in other types of gradient estimators.  Remarkably, our analysis (Theorem 3.2) identifies the optimal structure for these parameters, showing that a simple relation between them suffices to achieve optimal sample complexity. Therefore, we believe there still could be a large space to improve this approach.
>
>     * [Shamir2017] Shamir, Ohad. "An optimal algorithm for bandit and zero-order convex optimization with two-point feedback." *Journal of Machine Learning Research* 18.52 (2017): 1-11.
>     * [Duchi2015] Duchi, John C., et al. "Optimal rates for zero-order convex optimization: The power of two function evaluations." *IEEE Transactions on Information Theory* 61.5 (2015): 2788-2806.

---

> > ### Comment · Reviewer_5co9 · 2025-08-03
> > **Good!**
> >
> > I'm satisfied with the proposed  corrections and save my score. Good luck!

---

### Official Review · Reviewer_sGPT · 2025-07-02

**Clarity:** 3
**Significance:** 3
**Originality:** 3
**Rating:** 4
**Confidence:** 4

**Summary:**

The authors considered the problem of zeroth-order optimization (ZOO) in a stochastic optimization setup. They argued that most existing ZOO methods have a biased estimate of the gradient. Therefore, they proposed a family of unbiased estimators of the gradient based on merely function evaluations. In particular, they constructed four unbiased estimators that depend on 1 to 4 function evaluations in estimating the gradient. The main idea in devising these estimators is to write $\nabla_v f(x)$ in a direction of vector $v$ as a telescoping series of gradient estimators with diminishing perturbation stepsize $\mu_n$. This formulation provides an unbiased estimator given in (6) (which is called $\text{P}_4$ estimator). Moreover, they suggested three other estimators with a reduced number of function evaluations. Under some regularity assumption on the objective function, they showed that the proposed family of the estimator is unbiased. Moreover, they gave upper bounds on the variance of gradient estimators in terms of perturbation step sizes $\{\mu_n\}$ and sampling distribution $\{p_n\}$. In particular, for the estimator with one evaluation function, the variance might be unbounded, but for the rest of the estimators, by tuning $\mu_n$ and $p_n$, the variances match the optimal order of classical two-point estimators. Moreover, for ease of implementation, they proposed a method that, given a sampling distribution, derived the perturbation stepsize, and they gave a formulation for the perturbation step size for two distributions of Geometric and Zipf distributions. The experimental results showed improvement over some previous work on synthetic functions and training of a language model.

**Questions:**

- What is the motivation behind having an unbiased estimator of the gradient? For instance, in the literature of variance reduction methods in stochastic optimization, most proposed methods are biased estimators while they reduce the variance. Overall, they can achieve the best known sample complexity while the estimator is biased.

- Compared to biased estimators in the ZOO setup, what are the advantages of the current work in terms of sample complexity? It is good to clarify any advantages in the paper.

- The explanation in lines 89-97 is not so clear to me. In particular, would you please explain more about ``the outcomes of a random variable have no naturally given order (which makes it different from a series), which requires a random variable’s expectation 95 to be well-defined regardless of any such ordering''. It seems that the assumption in Proposition 2.1 is needed because of this argument, which limits the applicability of the current work.

- Would you please compare the current method with (Chen 2020) in Figures 1 and 2? Moreover, it is also good to add optimization algorithms such as SGD and Adam in these figures as a reference.

- Could you elaborate on the impact of sampling distribution on the performance of the estimator or the optimization algorithm in general? In particular, in what cases, which one of the two considered sampling distributions should be used?

**Ethical Concerns:**

["NO or VERY MINOR ethics concerns only"]

**Final Justification:**

Overall, I consider this a good work that can inspire further improvements by other researchers in the area of zeroth-order methods. However, comparisons with (Chen, 2020), SGD, and Adam—particularly with (Chen, 2020)—should be included in the revised version. Regarding the assumption of second-order continuous differentiability, I still believe this is a strong condition that may not hold in many practical applications. Therefore, I would like to keep my score unchanged.

**Limitations:**

The assumptions are mentioned in the paper, but it is good to explain what the impacts of these assumptions are and for what applications the proposed estimators can be used. It seems that there was no discussion on the potential negative societal impact of the work. It is good to add any societal impacts or otherwise, mention that there is no immediate impact.

**Paper Formatting Concerns:**

No major formatting issue is noticed.

**Quality:**

3

**Strengths And Weaknesses:**

### Strengths
- The authors proposed a family of unbiased estimators of the gradient in the ZOO setting. It seems that there is only one unbiased estimator in the literature by (Chen 2020), which was a special case of $\text{P}_4$ estimator.

- The authors provided theoretical guarantees on the unbiasedness of the proposed family of estimators and also gave upper bounds on the variance for the three estimators mentioned in Section 2.3.

- They derived optimal sampling distributions for given perturbation step sizes that minimize the upper bound on the variance (Theorem 3.2). They also provided an efficient implementation of the estimator by fixing first the sampling distribution and then adjusting the perturbation step sizes accordingly.

### Weaknesses:
- The motivation behind the unbiased estimator is not well explained. In fact, it is not clear what the advantages of having an unbiased estimator in the ZOO setup are.

- The assumption on the objective function being a second-order continuously differentiable function is not well motivated, and it is not clear in which theorems this assumption is needed. Apparently, it is needed in all theorems and propositions, but it is not explicitly mentioned in the statements. I think this assumption is also restrictive, as neural networks with ReLU activation may not satisfy this assumption.

- In the experiments, it seems that the authors did not compare with the closest related work of (Chen 2020).

---

> ### Author Rebuttal · Authors · 2025-07-27
>
> We deeply appreciate the reviewer’s positive evaluation and constructive feedback. Below, we provide our point-by-point responses to each of the concerns raised.
>
> ## Response to questions
> * (**Motivation behind having an unbiased estimator of the gradient**) What is the motivation behind having an unbiased estimator of the gradient? For instance, in the literature of variance reduction methods in stochastic optimization, most proposed methods are biased estimators while they reduce the variance. Overall, they can achieve the best known sample complexity while the estimator is biased.
>
>     **Response:**  We thank the reviewer for raising this important question. We highlight two key advantages and one potential advantage of using an unbiased estimator:
>
>     (i) *Practical benefit:* It removes the need to balance bias and variance as a coupled trade-off. This simplifies the hyperparameter tuning process, allowing us to focus solely on controlling variance.  As indicated by our Theorem 3.2, the variance upper bounds of the $P_3$ and $P_4$ estimators are parameter-agnostic; that is, they do not introduce any additional hyperparameters compared to classical biased estimators.
>
>     (ii) *Theoretical benefit:* The absence of bias enables the direct application of standard optimization theory, which leads to tighter convergence guarantees without needing to account for the complications introduced by estimator bias. While our current analysis focuses on vanilla SGD (as in Corollary 3.5), the same theoretical framework readily extends to a broad range of optimization methods.
>
>     (iii) *Potential benefit:* The unbiased estimator opens a novel and underexplored direction in the design of zeroth-order gradient estimators. While existing (typically biased) estimators have been extensively studied and refined, the space of unbiased estimators remains largely untapped. We believe our work creates new opportunities for developing more effective and theoretically grounded zeroth-order estimators. For example, in our Proposition 2.1, we identify what kinds of assumptions are needed to represent a *Taylor series* as an expectation; this result could be extended to many other series and may motivate new zeroth-order estimators that can be tailored to the specific structure of the objective function.
>
> * (**Advantage in terms of sample complexity**) Compared to biased estimators in the ZOO setup, what are the advantages of the current work in terms of sample complexity? It is good to clarify any advantages in the paper.
>
>     **Response:** Corollary 3.5 shows that SGD with $P_3$/$P_4$-estimator achieves $\mathcal{O}(d/\epsilon^4)$ oracle complexity. This complexity is information‑theoretic optimal (that is, it doesn't depend on any specific form of the gradient estimator) and cannot be further improved (see Proposition 1 [Duchi et al. 2015]). From this perspective, all existing zeroth-order gradient estimators cannot outperform our proposed method in terms of sample complexity.
>
>     * [Duchi et al. 2015] Duchi, John C., et al. "Optimal rates for zero-order convex optimization: The power of two function evaluations." IEEE Transactions on Information Theory 61.5 (2015): 2788-2806.
>
> * (**Explain  lines 89-97**) The explanation in lines 89-97 is not so clear to me. In particular, would you please explain more about ``the outcomes of a random variable have no naturally given order (which makes it different from a series), which requires a random variable’s expectation 95 to be well-defined regardless of any such ordering''. It seems that the assumption in Proposition 2.1 is needed because of this argument, which limits the applicability of the current work.
>
>     **Response:** Thank you for the insightful comment. We will revise lines 89–97 to clarify the underlying idea. The goal of this paragraph is to explain when an infinite series can be interpreted as the expectation of a random variable. Consider the convergent series $\sum_{i=1}^\infty p_i x_i$.
>
>     1. What is the value of this infinite series?
>
>         We can calculate the finite-sum $S_n := \sum_{i=1}^n p_i x_i$; then we let $n$ tend to infinite.
>
>     2. What is the value of an expectation of the random variable?
>
>         We consider the random variable $X$ with $P(X=x_i) = p_i$. Then the expectation $E[X]$ can of course be written as  $\sum_{i=1}^\infty p_i x_i$. However, the notion of expectation must be well-defined independently of any ordering of outcomes (that is, **"regardless of any such ordering"**). That is, for an arbitrary permutation $\sigma: \mathbb{N}  \to \mathbb{N} $, all series   $\sum_{i=1}^\infty p_{\sigma(i)} x_{\sigma(i)}$ should represent the same expectation $E[X]$.
>
>     This invariance under reordering is guaranteed only when the series is absolutely convergent. Proposition 2.1 provides a sufficient condition to ensure this property. While the condition itself may appear stronger than absolute convergence, it has the advantage of being easier to verify in practice.
>
> * (**Compare (Chen 2020), SGD, and Adam**) Would you please compare the current method with (Chen 2020) in Figures 1 and 2? Moreover, it is also good to add optimization algorithms such as SGD and Adam in these figures as a reference.
>
>     **Response:** Thank you for the suggestion. Yes, we will include all three methods in Figure 2 to facilitate a direct comparison, as requested. Regarding Figure 1, it is designed specifically to evaluate the gradient estimation error and does not depend on any particular optimization algorithm. Therefore, in Figure 1, we will include a comparison with (Chen, 2020), but not with SGD or Adam, as they are not directly relevant in this context.
>
> * (**Impact of sampling distribution on the performance**) Could you elaborate on the impact of sampling distribution on the performance of the estimator or the optimization algorithm in general? In particular, in what cases, which one of the two considered sampling distributions should be used?
>
>     **Response:** Our theoretical analysis (Theorem 3.2) shows that both sampling distributions lead to estimators with optimal variance. This result is, in our view, quite surprising at the first glance. Because, intuitively, using the $P_3$ or $P_4$ estimator seems challenging in practice due to the need to tune an infinite-dimensional hyperparameter $\{ p_n \}$ and $\{ \mu_n \}$. However, Theorem 3.2 reveals a key insight: optimal complexity can still be achieved as long as these weights satisfy a simple algebraic relation.
>
>     To demonstrate this, we have provided an example based on the Zipf distribution, which illustrates how to construct $P_3$ or $P_4$ estimators when a closed-form solution is not available. Our empirical comparison between the Zipf and Geometric estimators in Figure 2 further supports Theorem 3.2, showing that their performances are indeed comparable, with no significant difference observed in practice.
>
>     Therefore, Zipf, Geometric, or otherwise, does not fundamentally affect the convergence rate or the variance of the estimator. All of them will achieve the optimal sample complexity.
>
>
> ## Response to weaknesses
>
> * (**The second-order continuously differentiable sssumption**) The assumption on the objective function being a second-order continuously differentiable function is not well motivated, and it is not clear in which theorems this assumption is needed. Apparently, it is needed in all theorems and propositions, but it is not explicitly mentioned in the statements. I think this assumption is also restrictive, as neural networks with ReLU activation may not satisfy this assumption.
>
>     **Response:** We thank the reviewer for this insightful comment. We would like to clarify the motivation for assuming second-order continuous differentiability. This assumption is introduced because our theoretical analysis relies on applying Taylor’s theorem to the second order, which requires the existence of Hessian $\nabla^2 f$. In contrast, assuming only Lipschitz continuity of the gradient (i.e., $L$-smoothness) is insufficient to guarantee this existence.
>
>     We acknowledge that this assumption excludes certain widely used neural network architectures, such as those employing ReLU activations (we also note that the $L$-smoothness assumption will also exclude this example). However, we would like to emphasize that this assumption is in line with prior work in zeroth-order optimization.  For example, the MeZO paper [MeZO] relies on the existence of the Hessian in its proof, even though they do not state this assumption explicitly.
>
>     * [MeZO] Malladi, Sadhika, et al. "Fine-tuning language models with just forward passes." *Advances in Neural Information Processing Systems* 36 (2023): 53038-53075.

---

> > ### Comment · Reviewer_sGPT · 2025-08-02
> >
> > I thank the authors for their response, which addressed most of my comments, with the exception of the comparison with (Chen, 2020), SGD, and Adam, particularly (Chen, 2020). Regarding the assumption of second-order continuous differentiability, I still believe this is a strong condition that may not hold in many applications. Overall, I consider this as a good work that can motivate further improvements by other researchers. Therefore, I would like to keep my score unchanged.

---

### Note · Authors · 2025-08-15

Thank you for the thoughtful reviews and for engaging during the discussion period. Below is a concise summary of the post-discussion outcome and key points from each reviewer. After rebuttal and discussion, our work received strong and broad support from the reviewers:
* `Reviewer 5co9`: Maintained a high score of **5 (Accept)** and confirmed being "satisfied with the proposed corrections".
* `Reviewer sGPT`: Maintained the **positive score** "unchanged" after we addressed unbiasedness motivation, Chen (2020) comparison, and notation refinements, explicitly showing the continued support for our work.
* `Reviewer nG3r`: Initially expressed concerns about comparison with the Forward AD method; after detailed discussion on Forward AD limitations (e.g., JVP unavailability in FlashAttention, theoretical memory usage), indicated they were "seriously considering increasing my score".
* `Reviewer Ea9x`:Stated that our rebuttal **"addresses most of my concerns"** and **"raise my score"** after our novelty clarifications, explanation of batch-size alignment in function-call plots, and presentation of additional comparison results.
* `Reviewer Yvd5`:  **Raised score from a 3 to a 5** after our clarification on how the selection random variable enables an unbiased four-evaluation estimator.

We sincerely appreciate the constructive feedback and support provided by the reviewers and the Area Chair throughout the review process.

Sincerely,

The Authors

---

### Decision · Program_Chairs · 2025-09-17

**Decision:**

Accept (spotlight)

**Comment:**

This paper proposes a family of unbiased zeroth-order gradient estimators based on a telescoping series representation of directional derivatives. The authors establish the conditions under which unbiasedness is achievable, derive optimal perturbation stepsizes and sampling distributions, and prove that their P3/P4 estimators achieve the optimal  complexity for smooth non-convex optimization. They show the minimal number of function evaluations for unbiased estimation can be reduced from 4 to 1. The rebuttal effectively addresses concerns: unbiasedness is achieved via randomization over an infinite sequence but w/o requiring infinite evaluations, and the method remains practical due to constraints. Most reviewers have raised their scores after discussion, confirming its merit. I recommend acceptance.